# LLM Performance Predictors are good initializers for Architecture Search

## Abstract

Large language models (LLMs) have become an integral component in solving a wide range of NLP tasks. In this work, we explore a novel use case of using LLMs to build performance predictors (PP): models that, given a specific deep neural network architecture, predict its performance on a downstream task. We design *PP prompts* for LLMs consisting of: (i) *role*: description of the role assigned to the LLM, (ii) *instructions*: set of instructions to be followed by the LLM to carry out performance prediction, (iii) *hyperparameters*: a definition of each architecture-specific hyperparameter and (iv) *demonstrations*: sample architectures along with their efficiency metrics and 'training from scratch' performance. For machine translation (MT) tasks, we discover that GPT-4 with our PP prompts (LLM-PP) can predict the performance of architecture with a mean absolute error matching the SOTA and a marginal degradation in rank correlation coefficient compared to SOTA performance predictors. Further, we show that the predictions from LLM-PP can be distilled to a small regression model (LLM-Distill-PP). LLM-Distill-PP models surprisingly retain the performance of LLM-PP largely and can be a cost-effective alternative for heavy use cases of performance estimation. Specifically, for neural architecture search (NAS), we propose a *Hybrid-Search* algorithm for NAS (HS-NAS), which uses LLM-Distill-PP for the initial part of search, resorting to the baseline predictor for rest of the search. We show that HS-NAS performs very similar to SOTA NAS across benchmarks, reduces search hours by ~50%, and in some cases, improves latency, GFLOPs, and model size.

## 1 Introduction

Large language models (LLMs) are useful in a wide range of tasks, which includes open-ended tasks (e.g., generation, brainstorming, and chat) and closed-ended tasks (e.g., summarization, question answering, and rewriting). In this work, we explore a novel use case of using LLMs: building a performance predictor (LLM-PP) for a deep neural network (DNN) architecture. The input to the predictor is the DNN architecture description, which is typically its hyperparameters (e.g., #layers, #attention heads per layer). The predictor outputs the performance (e.g., BLEU score (Papineni et al., 2002a)) of that architecture for a given downstream task. An ideal performance predictor (PP) should have low prediction errors (e.g., absolute difference) with respect to the performance obtained by training from scratch. We hypothesize that LLMs have a 'general understanding' of DNN architectures, which is learned from relevant training data sources such as DNN research papers and GitHub repositories. The main goal of this work is to unearth the architecture understanding capability of LLMs to design PPs that are: (i) accurate, (ii) efficient, and (iii) beneficial to other use cases (e.g., neural architecture search).

**How to design accurate performance predictor (PP)?** To answer this, we design *PP prompts* that specify the PP task precisely. Specifically, the prompts contain: (i) *role*: high-level description of the role assigned to the LLM, (ii) *instructions*: set of instructions to explain the details of the task (e.g., downstream task, architecture, performance/efficiency metric), which the LLM is expected to follow, (iii) *hyperparameters*: a definition of each architecture-specific hyperparameter and (iv) *demonstrations*: a set of supervised examples for the PP task, where each example contains the architecture description (e.g., hyperparameters, FLOPs) and its performance on the downstream task (e.g., BLEU score). In this work, we primarily use GPT-4 (OpenAI, 2023b) as the LLM and machine translation on popular WMT datasets as our downstream tasks. We discover that GPT-4 with our

PP prompts (LLM-PP) can predict the performance of an architecture with a mean absolute error matching the SOTA and a marginally worse performance in rank correlation coefficient, compared to SOTA weight-sharing supernet based performance predictors (Wang et al., 2020; Jawahar et al., 2023b).

Given our choice of GPT-4 for the LLM, LLM-PP requires using the GPT-4 API for scoring each architecture, which makes LLM-PP prohibitively expensive to be applied for many use cases. One such use case is neural architecture search (NAS), where the goal is to find an architecture that has optimal performance for a given constraint on a given hardware. In NAS, PP is typically used for each constraint (e.g., latency $\leq$ 100ms) to score roughly 3,000 candidate architectures (Wang et al., 2020). The pricing of GPT-4 as of August 2023 is 0.03\$ per $1K$ tokens [1]. Assuming PP prompts takes roughly one-third of $1K$ tokens, the estimated cost can be $\sim$30\$ for a single constraint on the target hardware. With different constraint values (e.g., 100ms, 200ms), constraint types (e.g., latency, FLOPs, memory), target hardwares (e.g., Nvidia A100, Raspberry Pi), the total cost can quickly become exorbitant (e.g., 1,800\$).

**How to design cost-effective PP?** To answer this, we distill the performance predictions of LLM-PP into a multilayer perceptron (MLP) based regression model (LLM-Distill-PP), while taking the architecture description (e.g., list of hyperparameters) as input features. Surprisingly, we find that LLM-Distill-PP can largely retain the performance of LLM-PP. Assuming LLM-Distill-PP needs only 3,000 examples, the estimated cost can be $\sim$30\$ for a single downstream task, which is amortized across different constraint values, constraint types, and target hardwares.

**Can LLM-Distill-PP speed up architecture search, while maintaining the efficiency and the quality of SOTA NAS?** To answer this, we use LLM-Distill-PP as PP for designing efficient MT architectures via SOTA NAS methods such as HAT (Wang et al., 2020) and Mixture-of-Supernets (Jawahar et al., 2023b). We propose *Hybrid-Search* search algorithm (HS-NAS) where LLM-Distill-PP is used as PP for the first half of the search budget (i.e., 15 iterations) and a weight-sharing supernet (SOTA performance predictor) is used as PP for the remaining 15 search iterations. HS-NAS is roughly 50% faster than SOTA NAS search, while performing similarly to (or improving on) architecture designed by SOTA NAS, and in some cases, enjoying reduced latency ($\sim$2%), FLOPs ($\sim$1%), and smaller model size ($\sim$2%).

**Main contributions:** (1) We propose LLM-PP, which uses few-shot prompting of LLM to build accurate performance predictors, achieving SOTA mean absolute error. (2) We further build LLM-Distill-PP, which naturally enjoys a better amortized cost than LLM-PP and is applicable for PP-heavy use cases (3) We introduce HS-NAS, a search algorithm that cuts the NAS search time by half compared to SOTA and identifies better efficient architectures, by exploiting advantages of LLM-Distill-PP and SOTA performance estimators. (4) We share the prompts, data used to train and evaluate LLM-Distill-PP models, alongside the code with detailed instructions for reproducibility.

## 2 RELATED WORK

**Performance Predictors.** A popular approach in NLP to build performance predictors is to train a weight-sharing supernet model that jointly trains a collection of architectures by sharing their weights with the largest model from the given search space (Wang et al., 2020; Yin et al., 2021; Xu et al., 2022a; Jawahar et al., 2023a;b). In each training step, an architecture is randomly sampled from the search space, weights corresponding to that architecture are extracted from the corresponding rows/columns from the weight matrices of the largest model and those weights are trained for the task of interest. After training, the performance of an architecture can be predicted by extracting weights corresponding to that architecture and measuring on the validation set of the task. The main challenges in supernet training include: (i) weight co-adaptation (Bender et al., 2018; Zhao et al., 2021), (ii) capacity bottleneck (Jawahar et al., 2023b), and (iii) gradient conflict (Gong et al., 2021).

**NAS for NLP.** Neural architecture search (NAS) is a general framework used to design efficient NLP architectures that satisfy user-defined constraints. The generality of NAS spans the following key dimensions: (i) *architecture family*: encoder-only (Yin et al., 2021; Xu et al., 2022a; 2021; 2022b), decoder-only (Javaheripi et al., 2022), encoder-decoder (Wang et al., 2020; Jawahar et al., 2023a;b) without restricting to Transformers, (ii) *constraint types*: latency, FLOPs, model size, and

---

[1] https://openai.com/pricing

(iii) *tasks*: task-agnostic pretraining (Xu et al., 2022a; Javaheripi et al., 2022; Jawahar et al., 2023b), task-specific training (Wang et al., 2020; Jawahar et al., 2023a). The search algorithm is generally based on evolutionary search, which uses a performance predictor to identify architectures of high quality. The algorithm also uses real or predicted efficiency metric to discard architectures that do not satisfy user defined efficiency constraint from the progressively evolving search space.

**LLMs for NAS.** GENIUS (Zheng et al., 2023) is a recent search algorithm that uses LLMs to generate candidate convolution based architectures for image classification tasks. GENIUS trains the candidate architecture from scratch to convergence to measure its performance, which can be prohibitively expensive for practical tasks. The key differences between GENIUS and our work are: (i) *LLM use case*: while GENIUS uses LLMs to *generate* candidate architectures, we use LLMs to *predict* performance of architecture, (ii) *search cost*: while GENIUS requires 8 NVIDIA V100 GPUs for training candidate architectures and roughly takes 5 days, the search cost for our work is upper bounded by SOTA NAS for MT, which is 1 NVIDIA V100 GPU and under 5 hours (Jawahar et al., 2023b), and (iii) *DNN backbone*: while GENIUS focuses on convolution based encoder-only architecture for image classification task, our work focuses on Transformer based encoder-decoder architecture for machine translation task. See Tornede et al. (2023) for a thorough survey on how LLM and AutoML (parent field of NAS) fields can reinforce each other. The background for other relevant topics such as LLMs and distilling LLMs can be found in A.1.

## 3 PERFORMANCE PREDICTION PROBLEM

We next formally define the performance prediction problem. Let $T$ denote a downstream task and $\mathcal{A}_T$ the corresponding search space of architectures. Let $\mathcal{Y}_T \subset \mathcal{R}$ denote the real space of performance scores for the task $T$. Let $\mathcal{D}_T$ denote the data distribution defined over $\mathcal{A}_T \times \mathcal{Y}_T$. Then, the performance predictor can be denoted by $f_T : \mathcal{A}_T \rightarrow \mathcal{Y}_T$. Let $\mathcal{L}_T^{test} = \{(\mathbf{a}_i, p_i)\}_{i=1}^m \sim (\mathcal{D})_T^m$, consisting of architecture, performance pairs $(\mathbf{a}_i, p_i)$ drawn i.i.d. from $\mathcal{D}_T$, denote the labeled test set. Note that $p_i$ is obtained by training the architecture $\mathbf{a}_i$ from scratch to convergence on task $T$, which we refer to as 'training from scratch' (TFS) performance. The quality of performance predictor can be gauged using two metrics: Mean absolute error (MAE) computes the mean of the absolute difference between predictions and their corresponding TFS performances, which can be formalized as $\sum_{(\mathbf{a}_i, p_i) \sim (\mathcal{D})_T} \frac{|f_T(\mathbf{a}_i) - p_i|}{|(\mathcal{D})_T|}$. Kendall rank correlation coefficient is another metric that computes the ranking correlation between a set of predictions and their corresponding TFS performances, and can be formalized as Kendall-Tau$([f_T(\mathbf{a}_1), \ldots, f_T(\mathbf{a}_m)], [p_1, \ldots, p_m])$. Recently, Jawahar et al. (2023b) showed that both MAE and Kendall-Tau metrics are crucial for evaluating the performance predictor quality. For instance, a predictor with 38% better MAE and 12% worse Kendall-Tau compared to a base predictor can power NAS to find an architecture that enjoys 4% BLEU improvement. On the other hand, a predictor with 5% worse MAE and 6% higher Kendall-Tau compared to a base predictor can lead to a NAS architecture that enjoys 0.1% BLEU improvement.

## 4 BASELINE PERFORMANCE PREDICTORS

The SOTA approach for building performance predictors ($f_T$) is to train a weight-sharing supernet model on the task $T$. We formalize the training objective of the supernet. Let the training data distribution be denoted $\mathcal{X}_{train}$. Let the training sample and label be denoted by $x, y$, where $x, y \sim \mathcal{X}_{train}$. Let the architecture sampled uniformly from the search space $\mathcal{A}_T$ be denoted by $a_{rand}$. Let $a_{large}$ and $a_{small}$ denote the largest and smallest architecture from the search space $\mathcal{A}_T$. Let the subnet with architecture $a$ be denoted by $s_a$. Let $s$ be parameterized by the supernet model weights $W$. The training objective of the supernet using sandwich sampling (Yu et al., 2020) is given by

$$\min_W \mathbb{E}_{x,y \sim \mathcal{X}_{train}}[\mathbb{E}_{a_{rand} \sim \mathcal{A}}[\mathcal{L}(s_{a_{rand}}(x; W), y)] \ + \ \mathcal{L}(s_{a_{large}}(x; W), y) \ + \ \mathcal{L}(s_{a_{small}}(x; W), y)].$$

Hardware-aware Transformers (Wang et al., 2020) uses single path one-shot (SPOS) optimization (Guo et al., 2020), where only $a_{rand}$ is optimized at every training step. Mixture-of-Supernets (Jawahar et al., 2023b) (MoS) uses mixture-of-experts (MoE) (Fedus et al., 2022) to increase the capacity of the supernet, and the router specializes the weights for each architecture. MoS was proposed in two variants: layer-wise MoS and neuron-wise MoS, which differ in the degree of freedom for weight generation. MoS uses sandwich sampling to train the supernet.

# 5 LLM PERFORMANCE PREDICTOR (LLM-PP)

LLM exhibits "general understanding" of DNN architectures, which is likely obtained by training on relevant data sources that describe DNN architectures such as research papers and GitHub repositories. These architecture understanding capabilities can be tested by prompting LLM to generate definition of hyperparameters, and generate design principles for architecture search (Zheng et al., 2023). For performance prediction, these capabilities can in turn help the LLM in mapping the DNN architectures to their performances well. To this end, we propose LLM based Performance Predictor (LLM-PP), which introduces the general idea of prompting an LLM to generate performance predictions for DNN architectures. The prompts, which we call as *PP prompts*, need to be carefully designed to communicate the performance prediction task precisely to the LLM. As shown in Figure 1, PP prompts decompose the task into four main components: *role*, *instructions*, *hyperparameters*, and *demonstrations*, followed by the test architecture. The *role* component contains the high-level description of the role assigned to the LLM, including the mention of downstream task (e.g., machine translation) and performance metric (e.g., BLEU). The *instructions* component contains a set of five instructions, which explain the details of the downstream task, the DNN architecture, and model efficiency metrics. The first two instructions are specific to the downstream task, and specify the type of the task (e.g., machine translation), dataset

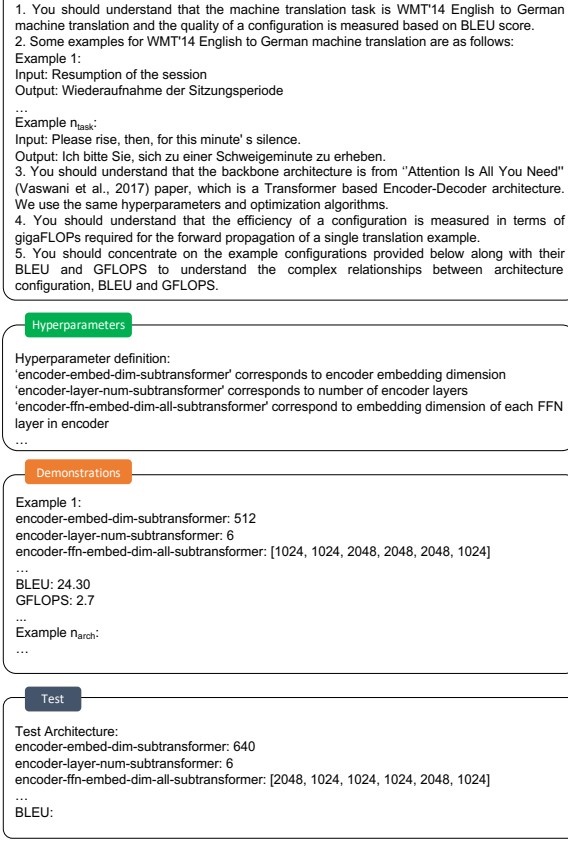

Figure 1: Prompt template to prompt LLM to generate performance predictions for WMT'14 EN-DE task. The expanded version of the prompt template can be seen in Appendix A.2.

(e.g., WMT'14 En-De), performance metric (e.g. BLEU), and inputs/outputs (e.g., source/target language) of $n_{task}$ examples from the dataset. The third instruction is specific to the DNN architecture, which includes architecture backbone (e.g., Transformer), type (e.g., encoder-decoder) and a reference to the original DNN paper. The fourth instruction contains the details of efficiency metric (e.g., GFLOPs), which will be included as part of demonstrations. The final instruction requires the LLM to condition its generation on the complex relationships between architecture configuration, performance, and efficiency metric. The third component of PP prompts, *hyperparameters*, contains the definition of each hyperparameter, which is specific to the architecture. *Demonstrations* is the final component which contains $n_{arch}$ supervised examples for the PP task. Each supervised example corresponds to an architecture sampled from the search space, with its hyperparameter values, efficiency score, and TFS performance score.

## 5.1 EVALUATION SETUP

**Downstream tasks.** For downstream tasks, we follow existing research (Wang et al., 2020; Jawahar et al., 2023a;b) and use the popular machine translation (MT) benchmarks: WMT'14 En-De, WMT'14 En-Fr, and WMT'19 En-De. The statistics of these benchmarks can be seen in A.4.1. We use BLEU (Papineni et al., 2002b) as the performance metric.

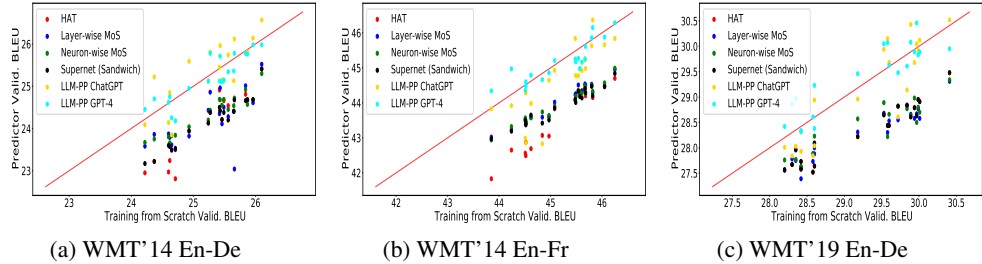

| | (a) WMT'14 En-De | (b) WMT'14 En-Fr | (c) WMT'19 En-De |

Figure 2: Training from scratch validation BLEU vs. performance predictor validation BLEU for WMT benchmarks. Performance scores from the optimal predictor should lie on the diagonal (red line). LLM-PP predicted performance scores are largely closer to the diagonal than other predictors.

| Dataset | WMT'14 En-De | | WMT'14 En-Fr | | WMT'19 En-De | | Average | |
|---|---|---|---|---|---|---|---|---|
| Performance Predictor | MAE | Kendall | MAE | Kendall | MAE | Kendall | MAE (↓) | Kendall (↑) |
| **Baseline** | | | | | | | | |
| HAT | 1.14 | 0.71 | 1.59 | 0.79 | 0.91 | 0.72 | 1.21 (0.00) | 0.74 (0.02) |
| Supernet (Sandwich) | 1.05 | **0.81** | 1.27 | 0.78 | 0.91 | 0.72 | 1.08 (0.00) | 0.77 (0.02) |
| Layer-wise MoS | 0.97 | 0.56 | 1.16 | 0.79 | 0.96 | **0.74** | 1.03 (0.01) | 0.70 (0.04) |
| Neuron-wise MoS | 0.87 | 0.79 | 1.18 | **0.87** | 0.87 | 0.67 | 0.97 (0.00) | **0.78** (0.01) |
| **LLM-PP** | | | | | | | | |
| ChatGPT | 0.42 | 0.52 | 0.82 | 0.61 | 0.72 | 0.56 | 0.65 (0.12) | 0.56 (0.04) |
| GPT-4 | 0.28 | 0.65 | **0.28** | 0.75 | 0.32 | 0.65 | **0.29** (0.00) | 0.68 (0.03) |
| **LLM-Distill-PP** | | | | | | | | |
| ChatGPT | 0.32 | 0.6 | 1.01 | 0.79 | 0.95 | 0.65 | 0.76 (0.09) | 0.68 (0.00) |
| GPT-4 | **0.22** | 0.64 | 0.34 | 0.76 | 0.38 | 0.68 | 0.31 (0.01) | 0.69 (0.01) |
| **LLM-PP GPT-4 Ablation** | | | | | | | | |
| Demonstraions only | 0.31 | 0.52 | 0.30 | 0.66 | 0.34 | 0.61 | 0.32 (0.02) | 0.60 (0.01) |
| + Role + Hyperparameters | 0.27 | 0.53 | 0.32 | 0.71 | 0.32 | 0.67 | 0.30 (0.01) | 0.64 (0.03) |
| + First instruction | 0.26 | 0.60 | 0.34 | 0.68 | 0.34 | 0.58 | 0.31 (0.02) | 0.62 (0.00) |
| + Second instruction | 0.27 | 0.60 | 0.31 | 0.72 | 0.35 | 0.66 | 0.31 (0.01) | 0.66 (0.02) |
| + Third instruction | 0.31 | 0.50 | 0.33 | 0.73 | **0.29** | 0.67 | 0.31 (0.01) | 0.63 (0.02) |
| + Fourth instruction | 0.25 | 0.63 | 0.32 | 0.65 | 0.33 | 0.71 | 0.30 (0.02) | 0.66 (0.03) |
| + Fifth instruction | 0.28 | 0.65 | **0.28** | 0.75 | 0.32 | 0.65 | **0.29** (0.00) | 0.68 (0.03) |

Table 1: Average MAE and Kendall-Tau between the performance predictor performance and the TFS performance, across three different seeds. The standard deviation is enclosed in parenthesis.

**DNN architecture.** We use the Transformer-based Encoder-Decoder architecture (Vaswani et al., 2017). The implementation and training settings as well as the search space ($\mathcal{A}$) are taken from Wang et al. (2020), which can be seen in A.4.2. The evaluation dataset (TFS-Eval) is taken from Jawahar et al. (2023b), which contains 30 architectures with their TFS performance scores for each WMT dataset. We use the implementation from Wang et al. (2020) to compute FLOPs, latency, and model size of architectures.

**Performance predictors.** The baseline performance predictors are as follows: (i) HAT (Wang et al., 2020), (ii) Supernet (Sandwich) (Jawahar et al., 2023b) (HAT, with sandwich sampling instead of SPOS), (iii) Layer-wise MoS (Jawahar et al., 2023b), and (iv) Neuron-wise MoS (Jawahar et al., 2023b). We build two LLM-PP variants, which differ in the choice of LLM: (i) ChatGPT (OpenAI, 2023a) (GPT-3.5-turbo, June version) and (ii) GPT-4 (OpenAI, 2023b) (June version) from OpenAI API. For PP prompts, we randomly sample: (i) 5 examples ($n_{task} = 5$) from the downstream task for the second instruction and (ii) 10 examples ($n_{task} = 10$) from TFS-eval for the demonstrations component. The remaining 20 examples from TFS-eval will be used for reporting the predictor quality. For all predictors, we repeat the experiments with three different seeds and report the average MAE and Kendall-Tau between the predictor performance and the TFS performance.

## 5.2 RESULTS

**LLM-PP predictions are closer to TFS performance scores than the baselines.** Figure 2 displays the training from scratch (TFS) versus performance predictor validation BLEU for different WMT benchmarks. The diagonal line (red line) corresponds to the perfect predictor, where the predicted performance exactly matches the TFS score. The predictions from the supernet based predictors (i.e.,

all non-LLM based ones) are clearly underestimates of the TFS performance for all architectures across three benchmarks. On the other hand, LLM-PP predictions are largely closer to the diagonal line, which showcases the high accuracy of LLM-PP.

**LLM-PP achieves SOTA MAE, while being marginally worse than baselines in Kendall-Tau.** The first and the second major rows of Table 1 show the MAE and Kendall-Tau of baseline and LLM-PP predictors. On average across datasets (last two columns), neuron-wise MoS is the best baseline, with the lowest MAE and highest Kendall-Tau score. LLM-PP ChatGPT and LLM-PP GPT-4 outperform Neuron-wise MoS in MAE, with LLM-PP GPT-4 achieving the SOTA MAE score. LLM-PP falls behind baselines marginally in terms of Kendall-Tau. In A.3, we inspect the histogram of distance between the items in the discordant pairs in the gold ranking for Neuron-wise MoS and LLM GPT-4. The discordant pairs of LLM-PP lie mostly around low gold ranking distances region (like Neuron-wise MoS), which should not ideally have a big negative impact for PP usecases (as seen in Section 7.3). The resulting CDF of gold ranking distances for discordant pairs for LLM-PP GPT-4 and Neuron-wise MoS are very similar. These results show that PP prompts can be used to design accurate performance predictors. Within LLM-PP, GPT-4 exceeds ChatGPT on both metrics across datasets.

**LLM-PP benefits from all the components of PP prompts.** The last major row of Table 1 shows the performance of ablating different components of PP prompts. LLM-PP's superior average performance stems from having all the PP prompt components together. Surprisingly, LLM-PP outperforms baselines in terms of MAE without any instructions (Demonstration only), which indicates the remarkable ability of LLM to pick up the performance prediction task just based on demonstrations. Although the MAE performance of different ablation variants is largely similar, the Kendall-Tau performance is different across variants. The second instruction which introduces downstream task specific examples and the fourth instruction which describes the efficiency metric are crucial for achieving high Kendall-Tau for LLM-PP.

# 6   DISTILLATION OF LLM-PP

Although LLM-PP achieves higher quality in performance prediction, the cost of LLM-PP increases linearly with the number of predictions. This cost can become exorbitant especially for high workload applications such as NAS, where the number of predictions is in several thousands. To illustrate the cost, we can look at the cost of NAS run by HAT (Wang et al., 2020) for a latency constraint on a given hardware, which requires evaluating roughly $3,000$ candidate architectures. The pricing of GPT-4 is 0.03\$ per 1K tokens, as of August 2023. Assuming PP prompts take roughly one-third of 1K tokens, the estimated cost will be roughly 30\$ ($\frac{0.03*3000}{3}$) for a single constraint on a given hardware. The total number of search runs depends directly on the number of constraint types (e.g., latency, memory, and FLOPs), values (e.g., 100ms, 200ms), and hardware (e.g., Nvidia A100, Raspberry Pi). If the number of constraint types is three, each constraint takes five possible values and there are four target hardwares, the estimated cost will become as high as roughly $1,800$\$ ($\frac{0.03*3000*3*5*4}{3}$) per downstream task. To build a cost-effective LLM-PP, we propose LLM-Distill-PP, which is trained on distilled outputs of LLM-PP. Specifically, LLM-Distill-PP is a multilayer perceptron based regressor, which is trained as follows: (1) A distillation dataset for the PP task is first created by sampling a few thousand architectures from the search space and recording the downstream task performance predicted by LLM-PP, (2) A regression model is trained using architecture-specific hyperparameters as features and the distilled output as label. Once trained, LLM-Distill-PP can be used to predict the performance of unseen architectures for the given downstream task. If the number of distillation examples is small (e.g., $3,000$), the estimated cost to query LLM-PP will be roughly 30\$ ($\frac{0.03*3000}{3}$). This one-time cost of LLM-Distill-PP is amortized across different constraint types, values, and hardwares (e.g., 60 search runs), thereby leading to $98.3\%$ (from $1,800$\$ to 30\$) reduction in cost.

**Setup.** The feature vector (or encoding) of each architecture is detailed in A.4.3. The hyperparameters of LLM-Distill-PP's regression model are borrowed from HAT's latency predictor: 3 hidden layers, 400 as hidden dimension, 128 as batch size, 1e-5 as learning rate, and 5000 as training steps. For each downstream task, we use only 3000 architecture examples to distill from LLM-PP.

**Results.** LLM-Distill-PP's results are shown in the third major row of Table 1. Despite the simple model design, LLM-Distill-PP can largely perform similarly or better than LLM-PP for both

ChatGPT and GPT-4. For ChatGPT, LLM-Distill-PP improves over LLM-PP on average MAE and Kendall-Tau by roughly 17%. For GPT-4, LLM-Distill-PP lags behind LLM-PP in average MAE by 7%, while enjoying similar Kendall-Tau. Impressively, LLM-Distill-PP achieves the SOTA MAE for WMT'14 En-De task, with an improvement over LLM-PP by 20%. The improvements of LLM-Distill-PP can be due to two factors. First is *regularization*, where LLM-Distill-PP's simple model design of using regression model of few layers outweighs the complicated modeling of LLM underlying LLM-PP. Another regularization aspect is due to LLM-Distill-PP's simplistic architecture-specific features. The second factor is LLM-Distill-PP's *context specialization* where few thousands of examples are used to follow the task. On the other hand, LLM-PP has to rely only on the components of PP prompt (including few demonstrations) and pretrained knowledge to follow the task.

# 7 LLM-DISTILL-PP FOR ARCHITECTURE SEARCH

---

**Algorithm 1** Hybrid-Search algorithm for Neural Architecture Search (HS-NAS). Changes to HAT (Wang et al., 2020)'s search algorithm are in red color.

---

**Input:** LLM-Distill-PP model: `llm-distill-pp`, Weight-sharing supernet: `supernet`, Latency predictor: `latency-predictor`, #Search iterations: `num-iterations`, Population size: `population-size`, #Parents: `num-parents`, #Mutations: `num-mutations`, #Crossovers: `num-crossover`, Mutate probability: `mutate-prob`, Latency constraint: `latency-constraint`, LLM-Distill-PP Start Iteration: `llm-start-iteration`, LLM-Distill-PP End Iteration: `llm-end-iteration`

**Output:** `best-architecture`

1: $popu \leftarrow$ `population-size` random samples from the search space // create init. population
2: **for** $iter \leftarrow 1$ to `num-iterations` **do**
3:     // generate parents by picking top candidate architectures
4:     **if** `llm-start-iteration` $< iter <$ `llm-end-iteration` **then**
5:         `parents` $\leftarrow$ top 'num-parents' architectures from $popu$ by `llm-distill-pp`
6:     **else**
7:         `parents` $\leftarrow$ top 'num-parents' architectures from $popu$ by `supernet`
8:     `mutate-popu` $= \{\}$ // generate candidates via mutation
9:     **for** $mi \leftarrow 1$ to `num-mutations` **do**
10:         `gene` $\leftarrow$ mutate a random example from $popu$ with `mutate-prob`
11:         **if** `gene` satisfies `latency-constraint` via `latency-predictor` **then**
12:             `mutate-popu` $=$ `mutate-popu` $\cup$ `gene`
13:     `crossover-popu` $= \{\}$ // generate candidates via cross-over
14:     **for** $ci \leftarrow 1$ to `num-crossover` **do**
15:         `gene` $\leftarrow$ crossover two random examples from $popu$
16:         **if** `gene` satisfies `latency-constraint` via `latency-predictor` **then**
17:             `crossover-popu` $=$ `crossover-popu` $\cup$ `gene`
18:     $popu =$ `parents` $\cup$ `mutate-popu` $\cup$ `crossover-popu` // update population
19: **return** top architecture from $popu$

---

Given that LLM-Distill-PP can achieve high performance prediction quality while being cost-effective, we study their application for a real world task: NAS. In NAS, performance predictors are typically used to rank a set of candidate architectures to identify high-performing architectures. As discussed in Section 2, existing NAS research for NLP primarily use weight-sharing supernet as a performance predictor. Hence, we study the interesting research question: *Can LLM-Distill-PP speed up architecture search, while maintaining the efficiency and the quality of SOTA NAS?* To this end, we propose the Hybrid-Search algorithm for NAS (HS-NAS), which will be detailed now.

## 7.1 HYBRID-SEARCH ALGORITHM FOR NAS

The key idea behind HS-NAS algorithm is to use the LLM-Distill-PP for subset of the search iterations, while resorting to supernet for the rest of the iterations. In this work, we apply this general idea on the evolutionary search algorithm proposed in HAT. The details of the HS-NAS algorithm is shown in Algorithm 1, where the changes to HAT's search algorithm are highlighted in red color. LLM-Distill-PP will be used as performance predictor for all the search it-

| Search Algorithm | BLEU (↑) | Latency (ms) (↓) | GFLOPs (↓) | Model Size (M) (↓) | Search Hours (↓) |
|---|---|---|---|---|---|
| **WMT'14 En-De** | | | | | |
| HAT | 27.9 | 102.0 | 3.0 | 64.4 | 1.09 |
| Layer-wise MoS | 27.8 | 100.4 | 3.08 | 64.4 | 1.45 |
| Neuron-wise MoS | **28.0** | **99.0** | 3.26 | 72.2 | 1.39 |
| HS-NAS (GPT-4, HAT, 1, 15) | 27.9 | 99.7 | **2.96** | **63.1** | **0.56** |
| **WMT'14 En-Fr** | | | | | |
| HAT | 40.8 | **96.4** | 2.61 | **63.8** | 6.33 |
| Layer-wise MoS | 40.5 | 99.4 | 2.96 | 70.5 | 6.81 |
| Neuron-wise MoS | **40.9** | 97.6 | 3.13 | 70.5 | 7.03 |
| HS-NAS (GPT-4, HAT, 1, 15) | 40.7 | 98.2 | **2.54** | **63.8** | **3.15** |
| **WMT'19 En-De** | | | | | |
| HAT | 44.7 | 100.8 | 3 | 73.06 | 1.11 |
| Layer-wise MoS | **44.9** | 96.8 | 3.26 | 82.95 | 1.13 |
| Neuron-wise MoS | **44.9** | 122.4 | 3.34 | 82.95 | 1.21 |
| HS-NAS (GPT-4, HAT, 1, 15) | 44.4 | **70.0** | **2.51** | **66.36** | **0.46** |

Table 2: HS-NAS versus SOTA NAS on three MT benchmarks for latency constraint of $100ms$ - Test BLEU, latency in milliseconds, GFLOPs, model size in millions, and search hours.

| Search Algorithm | BLEU (↑) | Latency (ms) (↓) | GFLOPs (↓) | Model Size (M) (↓) | Search Hours (↓) |
|---|---|---|---|---|---|
| **100ms** | | | | | |
| HAT | 40.8 | **96.4** | 2.61 | **63.8** | 6.33 |
| Layer-wise MoS | 40.5 | 99.4 | 2.96 | 70.5 | 6.81 |
| Neuron-wise MoS | **40.9** | 97.6 | 3.13 | 70.5 | 7.03 |
| HS-NAS (GPT-4, HAT, 1, 15) | 40.7 | 98.2 | **2.54** | **63.8** | **3.15** |
| **150ms** | | | | | |
| HAT | 41.3 | 176.4 | **3.31** | **74.3** | 7.33 |
| Layer-wise MoS | **41.4** | **158.7** | 4.3 | 92.8 | 8.39 |
| Neuron-wise MoS | **41.4** | 200.2 | 4.26 | 92.8 | 8.35 |
| HS-NAS (GPT-4, HAT, 1, 15) | **41.4** | 172.6 | **3.31** | **74.3** | **3.69** |
| **200ms** | | | | | |
| HAT | 41.5 | 187.5 | **3.7** | **79.5** | 7.8 |
| Layer-wise MoS | 41.4 | 205.6 | 4.49 | 99.4 | 8.63 |
| Neuron-wise MoS | 41.6 | **184.1** | 4.53 | 99.4 | 8.77 |
| HS-NAS (GPT-4, HAT, 1, 15) | **42.0** | 187.8 | **3.7** | **79.5** | **3.88** |

Table 3: HS-NAS versus SOTA NAS on WMT'14 En-Fr for different latency constraints - Test BLEU, latency in milliseconds, GFLOPs, model size in millions, and search hours.

erations in between `llm-start-iteration` and `llm-end-iteration`. In rest of the iterations, supernet will be used as performance predictor. When `llm-start-iteration=1` and `llm-end-iteration=num-iterations`, HS-NAS uses LLM-Distill-PP as performance predictor for all the search iterations. When `llm-start-iteration=-1` and `llm-end-iteration=-1`, HS-NAS applies supernet as performance predictor for all the search iterations, which is exactly HAT's original search algorithm. HS-NAS comes with four arguments: (`llm-distill-pp`, `supernet`, `llm-start-iteration`, `llm-end-iteration`).

## 7.2 SEARCH AND EVALUATION SETUP

For all our search experiments, we use LLM-Distill-PP GPT-4 as `llm-distill-pp` due to its superior performance over the ChatGPT counterpart (see the third major row in Table 1). The hyperparameters of HS-NAS's search algorithm are taken from HAT: `num-iterations=30`, `population-size=125`, `num-parents=25`, `num-mutations=50`, `num-crossover=50`, and `mutate-prob=0.3`. We experiment with three `latency-constraints`: $100ms$, $150ms$, and $200ms$. We use the `latency-predictor` and `supernet` from HAT. Once the search returns the best architecture, the final weights for this architecture is obtained by training the architecture from scratch to convergence using HAT's training settings (see A.4.2). The details of the target hardware, efficiency metric for search (search hours), and architecture (latency, GFLOPs, model size) can be seen in A.5.

| Search Algorithm | BLEU (↑) | Latency (ms) (↓) | GFLOPs (↓) | Model Size (M) (↓) | Search Hours (↓) |
|---|---|---|---|---|---|
| HAT | 27.9 | 102.0 | 3.0 | 64.4 | 1.09 |
| HS-NAS (GPT-4, HAT, 1, 30) | 27.5 | 99.3 | 3.34 | 72.2 | 0.04 |
| HS-NAS (GPT-4, HAT, 1, 5) | 27.4 | 100.4 | 2.96 | 63.1 | 0.97 |
| HS-NAS (GPT-4, HAT, 25, 30) | 28.0 | 119.1 | 3.18 | 70.9 | 0.95 |
| HS-NAS (GPT-4, HAT, 1, 15) | **27.9** | **99.7** | **2.96** | **63.1** | **0.56** |
| HS-NAS (GPT-4, HAT, 16, 30) | 27.6 | 101.7 | 3.34 | 72.2 | 0.75 |
| HS-NAS (GPT-4, HAT, 1, 25) | 27.7 | 98.9 | 3.01 | 63.1 | 0.23 |

Table 4: HS-NAS versus HAT on WMT'14 En-De for latency constraint: $100ms$ - Test BLEU, latency in milliseconds, GFLOPs, model size in millions, and search hours.

## 7.3 RESULTS

**Varying benchmarks.** HS-NAS performs largely similar to SOTA across benchmarks, reduces search hours by ~50% and in some cases, improves latency, GFLOPs, and model size. This trend is evident from Table 2 that shows the comparison of HS-NAS (GPT-4 as LLM-Distill-PP, HAT as supernet, 1 as LLM start iteration, 15 as LLM end iteration) and SOTA NAS for latency constraint of 100ms. This trend highlights that LLMs are good initializers for architecture search.

**Varying latency constraints.** HS-NAS's trend largely holds true across different latency constraints. Table 3 shows the comparison of the HS-NAS recipe (GPT-4, HAT, 1, 15) against SOTA NAS for various latency constraints: 100ms, 150ms, and 200ms. Besides reducing the search hours by 50%, HS-NAS achieves similar or better GFLOPs and same model size compared to SOTA NAS.

**Varying start and end iteration pairs.** Among different start and end iteration pairs, HS-NAS that uses LLM-Distill-PP GPT-4 for first 50% of iterations and HAT supernet for the rest, performs similarly or improves over HAT on all metrics. Table 4 shows the results of HS-NAS for various start and end iteration pairs. Using LLM-Distill-PP for all search iterations achieves lower performance, which indicates that marginal degradation in Kendall-Tau prevents LLM-Distill-PP from fully handling the search. These trends highlights that predictor having SOTA MAE scores seems useful for the first part of search, while predictor having SOTA Kendall-Tau seems useful for the rest of search.

**Varying initialization seeds, FLOPs constraints, underlying supernet.** HS-NAS seems robust to initialization effects caused by different seeds, achieving largely similar numbers on all metrics. This result is detailed in A.6.1. HS-NAS performs similarly to HAT for different FLOPs constraints, with at least 16% reduction in search hours, 1.2% improvement in latency, same GFLOPs and same model size. These trends largely hold true across benchmarks as well, as detailed in A.6.2. The dominance of HS-NAS seems consistent across the underlying supernet (second argument), as detailed in A.6.3.

**Trivially constructed efficient adaptations of SOTA** Search hours can be trivially reduced in several ways: halving the total number of search iterations and/or using distilled SOTA predictor instead of using supernet predictor directly. While these adaptations lead to a big drop in BLEU performance (1.8% for HAT (`num-iter.=15`)) or a big increase in latency and GFLOPs (9.7% and 32% respectively for Distilled HAT (`num-iter.=15`)), HS-NAS dominates these adaptions in search hour reductions, while maintaining SOTA performance and not degrading on any footprint metric, as detailed in A.6.4.

Putting all the observed trends of HS-NAS together, we find that the generality of HS-NAS extends to constraint types (latency, FLOPs), constraint values (different latencies, different FLOPs), different tasks (MT benchmarks), and underlying supernet (HAT, Neuron-wise MoS), while being robust to initialization effects.

## 8 CONCLUSION

In this work, we showed that LLM can be used to design accurate, cost-effective performance predictor, which also benefits neural architecture search. Our work contributes to the growing area of research in using LLM for NAS, with a new direction on building performance predictor. Future NAS research should explore the full potential of LLM by studying the application of LLM for candidate architecture generation and performance prediction jointly.

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

| Dataset | Year | Source Lang | Target Lang | #Train | #Valid | #Test |
|---------|------|-------------|-------------|--------|--------|-------|
| WMT | 2014 | English (en) | German (de) | 4.5M | 3000 | 3000 |
| WMT | 2019 | English (en) | German (de) | 43M | 2900 | 2900 |
| WMT | 2014 | English (en) | French (fr) | 35M | 26000 | 26000 |

Table 5: Statistics - Machine translation benchmark.

# A APPENDIX

## A.1 RELATED WORK - EXTENDED

**LLMs.** LLMs can be classified into two categories based on their training methods: foundation and instruction-tuned LLMs. Foundation LLMs, which includes GPT-3 (Brown et al., 2020), GLaM (Du et al., 2022), LLaMA-1 (Touvron et al., 2023a), undergo language model training on unannotated corpus from the web. These LLMs typically encode a lot of useful knowledge in their parameters and can be used for a downstream task by either fine-tuning or zero/few-shot prompting. Instruction-tuned LLMs are usually foundation LLMs that undergo instruction-tuning, where LLMs are explicitly fine-tuned to follow user defined instructions well. Such LLMs include Instruct-GPT (Ouyang et al., 2022), ChatGPT (OpenAI, 2023a), GPT-4 (OpenAI, 2023b), LLaMA-2 (Touvron et al., 2023b), and PaLM-2 (Anil et al., 2023). In practice, instruction-tuned LLMs can follow a wide range of user's instructions, even those that are outside the instruction tuning data distribution (Ouyang et al., 2022). However, depending on the task, instruction-tuned LLMs are prone to generating content that are factually incorrect, hallucinated, ignores user's instruction, toxic, and so on (Ouyang et al., 2022). These challenges make the current SOTA LLMs unreliable for critical applications such as medical diagnosis (Singhal et al., 2022).

**Distilling LLMs.** Distilling the generations from LLMs to smaller student models has become commonplace in NLP these days (Taori et al., 2023; Chiang et al., 2023; Wu et al., 2023; Mukherjee et al., 2023). The key motivations for such efforts include: (i) *cost reduction*: most LLMs are either behind a paywall or require high-end GPUs (e.g., NVIDIA A100) with high GPU memory (e.g., 80GB) to use, (ii) *latency reduction*: most LLMs are too slow even on high-end hardware (e.g., OPT-175B takes 4s for decoding 16 sequences of length 1024 on 8 NVIDIA A100 80GB GPUs (Xiao et al., 2022)), and (iii) *customization*: most LLMs are general purpose and are difficult to finetune. The commonly used distillation technique is sequence level knowledge distillation (Kim & Rush, 2016), where the student models are finetuned on responses from teacher LLMs via a standard language modeling objective.

## A.2 PROMPT TEMPLATE - EXPANDED VERSION

The expanded version of the prompt template can be seen in Figure 3.

## A.3 KENDALL-TAU - FINE-GRAINED ANALYSIS

We perform a fine-grained analysis of Kendall-Tau performance for Neuron-wise MoS and LLM-PP GPT-4. In figure 4, we plot the histogram of distance between the items in the discordant pairs in the gold ranking for Neuron-wise MoS and LLM GPT-4 across three MT benchmarks. The discordant pairs of LLM-PP lie mostly around low gold ranking distances region (like Neuron-wise MoS), which should not ideally have a big negative impact for the NAS task. In figure 5, we plot the corresponding cummulative distribution function (CDF). The CDF of gold ranking distances for discordant pairs for LLM-PP GPT-4 and Neuron-wise MoS are very similar.

## A.4 MACHINE TRANSLATION DETAILS

### A.4.1 MACHINE TRANSLATION - DATASET STATISTICS

The statistics of the MT benchmarks is shown in Table 5.

**Role**

You are a performance estimator for machine translation task, where you will estimate the BLEU score for the test architecture.

**Instruction**

You should follow these instructions:
1. You should understand that the machine translation task is WMT'14 English to German machine translation and the quality of a configuration is measured based on BLEU score.
2. Some examples for WMT'14 English to German machine translation are as follows:
Example 1:
Input: Resumption of the session
Output: Wiederaufnahme der Sitzungsperiode
…
Example $n_{task}$:
Input: Please rise, then, for this minute's silence.
Output: Ich bitte Sie, sich zu einer Schweigeminute zu erheben.
3. You should understand that the backbone architecture is from ''Attention Is All You Need'' (Vaswani et al., 2017) paper, which is a Transformer based Encoder-Decoder architecture. We use the same hyperparameters and optimization algorithms.
4. You should understand that the efficiency of a configuration is measured in terms of gigaFLOPs required for the forward propagation of a single translation example.
5. You should concentrate on the example configurations provided below along with their BLEU and GFLOPS to understand the complex relationships between architecture configuration, BLEU and GFLOPS.

**Hyperparameters**

Hyperparameter definition:
'encoder-embed-dim-subtransformer' corresponds to encoder embedding dimension
'encoder-layer-num-subtransformer' corresponds to number of encoder layers
'encoder-ffn-embed-dim-all-subtransformer' correspond to embedding dimension of each FFN layer in encoder
'encoder-self-attention-heads-all-subtransformer' correspond to number of self attention heads in each encoder layer
'decoder-embed-dim-subtransformer' corresponds to decoder embedding dimension
'decoder-layer-num-subtransformer' corresponds to number of decoder layers
'decoder-ffn-embed-dim-all-subtransformer' correspond to embedding dimension of each FFN layer in decoder
'decoder-self-attention-heads-all-subtransformer' correspond to number of self attention heads in each decoder layer
'decoder-ende-attention-heads-all-subtransformer' correspond to number of cross attention heads in each decoder layer
'decoder-arbitrary-ende-attn-all-subtransformer' correspond to number of encoder layers attended by cross-attention heads in each decoder layer (-1 means only attend to the last layer; 1 means attend to last two layers, 2 means attend to last three layers)

**Demonstrations**

Example 1:
encoder-embed-dim-subtransformer: 512
encoder-layer-num-subtransformer: 6
encoder-ffn-embed-dim-all-subtransformer: [1024, 1024, 2048, 2048, 2048, 1024]
encoder-self-attention-heads-all-subtransformer: [4, 8, 8, 8, 4, 4]
decoder-embed-dim-subtransformer: 512
decoder-layer-num-subtransformer: 4
decoder-ffn-embed-dim-all-subtransformer: [2048, 1024, 1024, 1024]
decoder-self-attention-heads-all-subtransformer: [4, 4, 8, 4]
decoder-ende-attention-heads-all-subtransformer: [4, 8, 8, 8]
decoder-arbitrary-ende-attn-all-subtransformer: [-1, -1, 1, -1]
BLEU: 24.30
GFLOPS: 2.7
...
Example $n_{arch}$:
…

**Test**

Test Architecture:
encoder-embed-dim-subtransformer: 640
encoder-layer-num-subtransformer: 6
encoder-ffn-embed-dim-all-subtransformer: [2048, 1024, 1024, 1024, 2048, 1024]
encoder-self-attention-heads-all-subtransformer: [4, 8, 8, 4, 4, 4]
decoder-embed-dim-subtransformer: 512
decoder-layer-num-subtransformer: 3
decoder-ffn-embed-dim-all-subtransformer: [1024, 2048, 2048]
decoder-self-attention-heads-all-subtransformer: [8, 8, 8]
decoder-ende-attention-heads-all-subtransformer: [8, 4, 4]
decoder-arbitrary-ende-attn-all-subtransformer: [-1, 1, 1]
BLEU:

Figure 3: Prompt template to prompt LLM to generate performance predictions for WMT'14 EN-DE task.

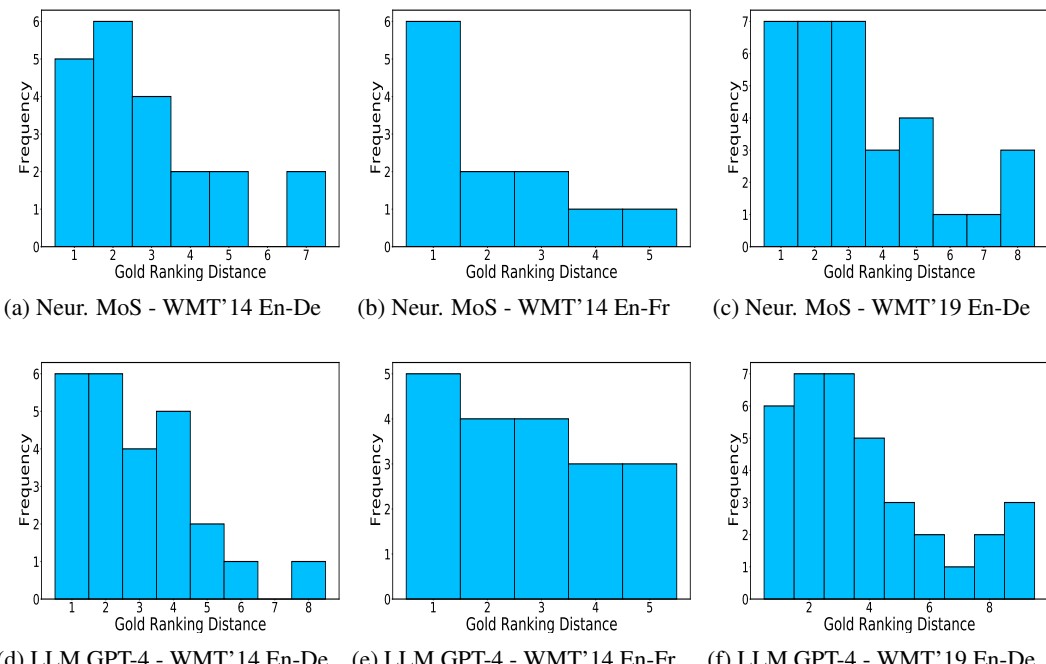

Figure 4: Histogram of distance between the items in the discordant pairs in the gold ranking for Neuron-wise MoS and LLM GPT-4 across three MT benchmarks. The discordant pairs of LLM-PP lie mostly around low gold ranking distances region (like Neuron-wise MoS), which should not ideally have a big negative impact for the NAS task.

### A.4.2 MACHINE TRANSLATION - TRAINING DETAILS AND SEARCH SPACE

Settings for training machine translation model include: $40K$ training steps, a cosine learning rate scheduler, Adam optimizer, and a warmup of learning rate from $10^{-7}$ to $10^{-3}$ with cosine annealing. The validation loss is used for model selection. The beam size is four with length penalty of 0.6. The search space ($\mathcal{A}$) is borrowed from HAT (Wang et al., 2020), which is also shown in Table 6.

### A.4.3 ARCHITECTURE ENCODING

Each machine translation architecture is encoded using a list of following 10 values:

1. *Encoder embedding dimension* corresponds to embedding dimension of the encoder.

2. *Encoder #layers* corresponds to number of encoder layers.

3. *Average encoder FFN. intermediate dimension* corresponds to average of FFN intermediate dimension across encoder layers.

4. *Average encoder self attention heads* corresponds to average of number of self attention heads across encoder layers.

5. *Decoder embedding dimension* corresponds to embedding dimension of the decoder.

6. *Decoder #Layers* corresponds to number of decoder layers.

7. *Average Decoder FFN. Intermediate Dimension* corresponds to average of FFN intermediate dimension across decoder layers.

8. *Average decoder self attention heads* corresponds to average of number of self attention heads across decoder layers.

9. *Average decoder cross attention heads* corresponds to average of number of cross attention heads across decoder layers.

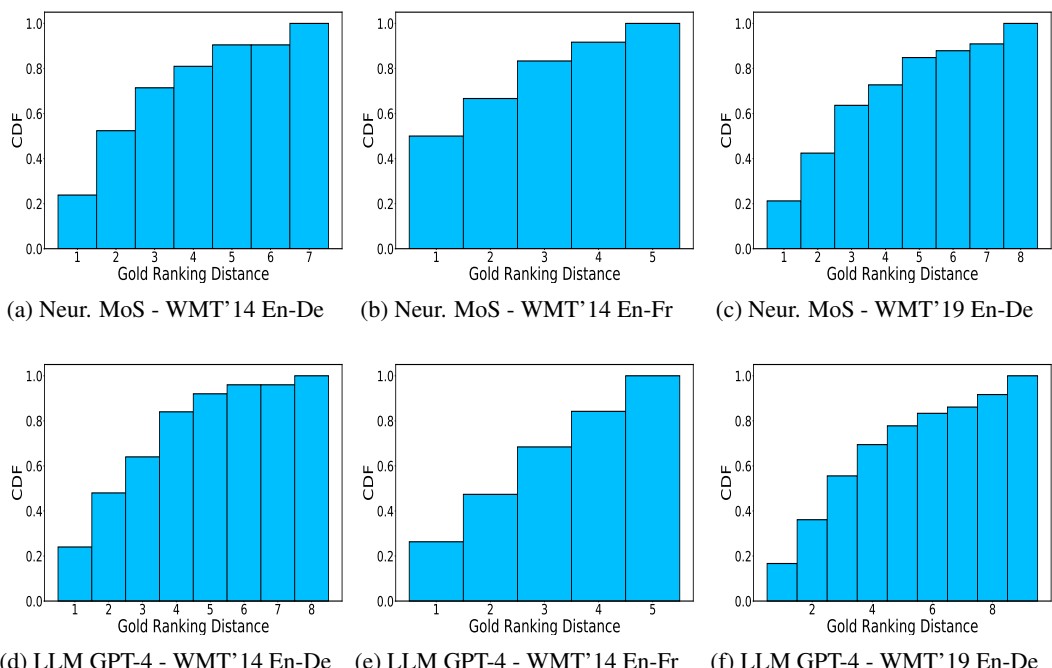

(a) Neur. MoS - WMT'14 En-De   (b) Neur. MoS - WMT'14 En-Fr   (c) Neur. MoS - WMT'19 En-De

(d) LLM GPT-4 - WMT'14 En-De   (e) LLM GPT-4 - WMT'14 En-Fr   (f) LLM GPT-4 - WMT'19 En-De

Figure 5: Cummulative distribution function of distance between the items in the discordant pairs in the gold ranking for Neuron-wise MoS and LLM GPT-4 across three MT benchmarks. The cummulative distribution function of gold ranking distances for discordant pairs for LLM-PP GPT-4 and Neuron-wise MoS are very similar.

| Hyperparameter Attribute | Value choices |
|---|---|
| Encoder-Embedding-Dim | {512, 640} |
| Decoder-Embedding-Dim | {512, 640} |
| #Encoder-Layers | {6} |
| #Decoder-Layers | {1, 2, 3, 4, 5, 6} |
| Encoder-QKV-Dim | {512} |
| Decoder-QKV-Dim | {512} |
| #Encoder-Self-Attention-Heads (PL) | {4, 8} |
| #Decoder-Self-Attention-Heads (PL) | {4, 8} |
| #Decoder-Cross-Attention-Heads (PL) | {4, 8} |
| #Decoder-Arbitrary-Attention (PL) | {-1, 1, 2} |
| Encoder-FFN-Intermediate-Embed-Dim (PL) | {1024, 2048, 3072} |
| Decoder-FFN-Intermediate-Embed-Dim (PL) | {1024, 2048, 3072} |

Table 6: Search space ($\mathcal{A}$), borrowed from HAT (Wang et al., 2020). 'PL' refers to hyperparameters that vary per layer.

10. *Average arbitrary encoder decoder attention* corresponds to average number of encoder layers attended by cross-attention heads in each decoder layer (-1 means only attend to the last layer, 1 means attend to the last two layers, 2 means attend to the last three layers).

## A.5 SEARCH AND EVALUATION SETUP - DETAILS

The target hardware for search is NVIDIA V100 GPU with 32GB GPU RAM. The efficiency metric for search is search hours, which accounts for the time taken to complete all the search iterations. We focus on the following architecture-specific efficiency metrics: (i) *latency* - time taken in milliseconds to encode a sentence in source language and generate the translation in target language, (ii) *GFLOPs* - gigaFLOPs taken for the feedforward propagation, and (iii) *model size* - number of architecture-specific parameters in millions. Scripts to compute these metrics are taken from HAT's

| Seed | BLEU (↑) | Latency (ms) (↓) | GFLOPs (↓) | Model Size (M) (↓) | Search Hours (↓) |
|---|---|---|---|---|---|
| **100ms** | | | | | |
| 1 | 40.7 | 104.1 | 2.54 | 63.8 | 3.14 |
| 2 | 40.7 | 98.2 | 2.54 | 63.8 | 3.15 |
| 3 | 40.7 | 101.2 | 2.58 | 63.8 | 3.16 |
| **150ms** | | | | | |
| 1 | 41.5 | 160.4 | 3.35 | 74.3 | 3.89 |
| 2 | 41.4 | 172.6 | 3.31 | 74.3 | 3.69 |
| 3 | 41.5 | 158.5 | 3.35 | 74.3 | 3.84 |

Table 7: Initialization effects of HS-NAS (GPT-4, HAT, 1, 15) on WMT'14 En-Fr for different latency constraints - Test BLEU, latency in milliseconds, GFLOPs, model size in millions, and search hours. HS-NAS seems robust to initialization effects, achieving similar numbers on all metrics of interest.

| Search | BLEU (↑) | Latency (ms) (↓) | GFLOPs (↓) | Model Size (M) (↓) | Search Hours (↓) |
|---|---|---|---|---|---|
| **2.5 GFLOPs** | | | | | |
| HAT | **26.9** | 69.5 | **2.47** | **41.0** | 2.54 |
| HS-NAS | 26.7 | **68.6** | **2.47** | **41.0** | **2.13** |
| **3.0 GFLOPs** | | | | | |
| HAT | 27.5 | 125.4 | **2.98** | **49.4** | 2.08 |
| HS-NAS | **27.6** | **123.9** | **2.98** | **49.4** | **1.51** |

Table 8: HS-NAS (GPT-4, HAT, 1, 15) vs. HAT on WMT'14 En-De for different FLOPs constraints - Test BLEU, latency in milliseconds, GFLOPs, model size in millions, and search hours. HS-NAS (GPT-4, HAT, 1, 15) performs similarly to HAT, with at least 16% reduction in search hours, 1.2% improvement in latency, same GFLOPs and same model size.

codebase [2] and we refer readers to the HAT paper for more details about how these metrics are computed.

## A.6 LLM-DISTILL-PP - EXTENDED RESULTS

### A.6.1 VARYING INITIALIZATION SEEDS.

HS-NAS seems robust to initialization effects caused by different seeds, achieving largely similar numbers on all metrics. This result is shown in Table 7, where latency numbers change slightly while numbers for other metrics are almost the same.

### A.6.2 VARYING FLOPS CONSTRAINTS.

HS-NAS performs similarly to HAT for different FLOPs constraints, with at least 16% reduction in search hours, 1.2% improvement in latency, same GFLOPs and same model size. Table 8 contains these superior results of HS-NAS across 2.5 and 3.0 GFLOPs constraints. These trends largely hold true across benchmarks as well, as shown in Table 9.

### A.6.3 VARYING UNDERLYING SUPERNET.

The dominance of HS-NAS seems consistent across the underlying supernet. In the results so far, HAT is the supernet used by HS-NAS. In Table 10, we replace HAT with Neuron-wise MoS and show that HS-NAS performs similarly to Neuron-wise MoS, with at least 50% reduction in search hours, better or similar model size and GFLOPs.

### A.6.4 TRIVIALLY CONSTRUCTED EFFICIENT ADAPTATIONS OF SOTA

Search hours can be trivially reduced in several ways: halving the total number of search iterations and/or using distilled SOTA predictor instead of using supernet predictor directly. As shown

---

[2] https://github.com/mit-han-lab/hardware-aware-transformers

| Search | BLEU (↑) | Latency (ms) (↓) | GFLOPs (↓) | Model Size (M) (↓) | Search Hours (↓) |
|---|---|---|---|---|---|
| **WMT'14 En-De** | | | | | |
| HAT | 27.5 | 125.4 | **2.98** | **49.4** | 2.08 |
| HS-NAS | **27.6 (+0.4%)** | **123.9 (-1.2%)** | **2.98** | **49.4** | **1.51 (-27.4%)** |
| **WMT'14 En-Fr** | | | | | |
| HAT | 39.4 | **69.6** | **2.99** | **49.1** | 6.69 |
| HS-NAS | **39.8 (+1%)** | 96.8 (+39.1%) | 3 | **49.1** | **4.2 (-37.2%)** |
| **WMT'19 En-De** | | | | | |
| HAT | 42.9 | 85.5 | **2.99** | **49.6** | 2.35 |
| HS-NAS | **43.1 (+0.5%)** | **71.9 (+15.9%)** | **2.99** | **49.6** | **2.03 (-13.6%)** |

Table 9: HS-NAS (GPT-4, HAT, 1, 15) vs. HAT across benchmarks for 3.0 GFLOPs constraint - Test BLEU, latency in milliseconds, GFLOPs, model size in millions, and search hours. HS-NAS (GPT-4, HAT, 1, 15) performs similarly or better than HAT, with at least 13% reduction in search hours, at least 1.2% improvement in latency (in most cases), same GFLOPs, and same model size.

| Search | BLEU (↑) | Latency (ms) (↓) | GFLOPs (↓) | Model Size (M) (↓) | Search Hours (↓) |
|---|---|---|---|---|---|
| **100ms** | | | | | |
| Neuron-wise MoS | **40.9** | **97.6** | **3.13** | **70.5** | 7.03 |
| HS-NAS (GPT-4, Neur., 1, 15) | **40.9** | 126.9 (+30%) | **3.13** | **70.5** | **3.36 (-52.2%)** |
| **150ms** | | | | | |
| Neuron-wise MoS | **41.4** | 200.2 | 4.26 | 92.8 | 8.35 |
| HS-NAS (GPT-4, Neur., 1, 15) | 41.3 (-0.2%) | **162.2 (19.0%)** | **4.22 (-0.9%)** | **91.5 (1.4%)** | **4.14 (-50.4%)** |
| **200ms** | | | | | |
| Neuron-wise MoS | 41.6 | **184.1** | **4.53** | **99.4** | 8.77 |
| HS-NAS (GPT-4, Neur., 1, 15) | **41.7 (+0.2%)** | 191.2 (+3.9%) | **4.53** | **99.4** | **4.22 (-51.8%)** |

Table 10: HS-NAS (GPT-4, Neuron-wise MoS, 1, 15) versus SOTA NAS on WMT'14 En-Fr for different latency constraints - Test BLEU, latency in milliseconds, GFLOPs, model size in millions, and search hours. HS-NAS is accompanied by four arguments: (`llm-distill-pp`, `supernet`, `llm-start-iteration`, `llm-end-iteration`). Across latency constraints, HS-NAS performs similarly or improves upon SOTA NAS, with at least 50% reduction in search hours, better or similar model size and GFLOPs.

| Search | BLEU (↑) | Latency (ms) (↓) | GFLOPs (↓) | Model Size (M) (↓) | Search Hours (↓) |
|---|---|---|---|---|---|
| HAT (`num-iter.=30`) | **27.9** | 102.0 | 3.0 | 64.4 | 1.09 |
| HAT (`num-iter.=15`) | 27.4 (-1.8%) | 107.6 (+5.5%) | **2.96 (-1.3%)** | **63.1 (-2%)** | 0.65 (-40.4%) |
| Distilled HAT (`num-iter.=15`) | 27.8 (-0.4%) | 111.9 (+9.7%) | 3.97 (+32%) | **63.1 (-2%)** | 0.58 (-46.8%) |
| HS-NAS (GPT-4, HAT, 1, 15) | **27.9** | **99.7 (-2.3%)** | **2.96 (-1.3%)** | **63.1 (-2%)** | **0.56 (-48.6%)** |

Table 11: HS-NAS versus trivial efficient adaptations of SOTA with half of the original search iterations (original `num-iterations` = 30): *original SOTA*, *distilled SOTA* on WMT'14 En-De for 100ms latency constraint - Test BLEU, latency in milliseconds, GFLOPs, model size in millions, and search hours. HS-NAS is accompanied by four arguments: (`llm-distill-pp`, `supernet`, `llm-start-iteration`, `llm-end-iteration`). Efficient adaptations of SOTA reduce search hours by at least 40%, at the expense of either a big drop in BLEU performance (1.8% for HAT (`num-iter.=15`) ) or big increase in latency and GFLOPs (9.7% and 32% respectively for Distilled HAT (`num-iter.=15`)). On the other hand, HS-NAS dominates these adaptions in search hour reductions, while maintaing the performance of SOTA and not degrading on any footprint metric.

in Table 11, the former approach suffers from a big drop in BLEU performance (1.8% for HAT (`num-iter.=15`)), while the latter approach suffers from a big increase in latency and GFLOPs (9.7% and 32% respectively for Distilled HAT (`num-iter.=15`)). On the other hand, HS-NAS dominates these adaptions in search hour reductions, while maintaining the performance of SOTA and not degrading on any footprint metric.

