# OpenReview forum: "LLM Performance Predictors are good initializers for Architecture Search"
_ICLR.cc/2024/Conference — Submitted to ICLR 2024_

### Official Review · Reviewer_ABPv · 2023-10-29

**Soundness:** 3 good
**Presentation:** 3 good
**Contribution:** 3 good
**Rating:** 6
**Confidence:** 4

**Summary:**

This paper explores the use of Large Language Models (LLMs) to build Performance Predictors (PP) for deep neural network architectures. These PP models aim to predict the performance of a given architecture on downstream tasks. The authors design PP prompts for LLMs, providing them with the role, instructions, architecture-specific hyperparameters, and demonstrations to guide the performance prediction process.

**Strengths:**

1. Applied LLM to Performance Prediction: The paper successfully applies Large Language Models (LLMs) to performance prediction for deep neural network architectures. This innovative use of LLMs for performance estimation can potentially benefit a wide range of applications in the field of deep learning.

2. Distillation Technology for Cost Reduction: The paper introduces a valuable approach to reduce the cost of using LLMs for performance prediction. The distillation process allows the transfer of knowledge from the LLM-PP models to smaller, more efficient regression models, making it a cost-effective alternative for performance estimation.

3. Hybrid Search Algorithm Accelerates Search Time: The Hybrid-Search algorithm (HS-NAS) presented in the paper demonstrates significant advantages in accelerating search time for Neural Architecture Search (NAS). It reduces search hours by approximately 50% and offers potential improvements in latency, GFLOPs, and model size. This can be a substantial advantage for practitioners looking to optimize their deep learning models.

4. Good Ablation Experiments: The paper conducts thorough ablation experiments to assess the effectiveness of their methods. This provides a clear understanding of the impact of different components and helps validate the proposed techniques.

**Weaknesses:**

1. Insufficient Innovation in Hybrid Search Algorithm: One potential drawback is the perceived lack of significant innovation in the Hybrid-Search algorithm. While it effectively accelerates search time, it may not introduce groundbreaking advancements in the field of NAS. More innovative aspects of the algorithm could enhance its contribution.

2. Fixed Downstream Tasks, Unknown Effects on Other Tasks: The paper primarily focuses on performance prediction for specific downstream tasks. However, it does not explore the potential impact or applicability of LLM-PP or LLM-Distill-PP models on a broader range of tasks. This limits the generalizability of the approach and its potential in different contexts.

**Questions:**

- Is the prediction result provided by LLM repeatable? How is it handled if the results given each time are different?
- The article mentioned that LLM  exhibits a "general understanding" of the DNN architectures.But how do you ensure that LLM understands the DNN framework rather than "reading memory" from its training data to provide prediction results?
- Since the final search should use the model distilled from LLM-PP instead of LLM-PP itself, why not use a PP that performs better than LLM-PP for distillation?

---

> ### Author Response · Authors · 2023-11-17
> **Response**
>
> Thanks for the careful review and for the helpful comments. We respond to quoted comments below:
> * **Weakness 1:** The hybrid search algorithm is based on a simple yet novel idea: combine the strengths of LLM and supernet based performance predictors. The simplicity in the algorithm design allows hybrid search algorithms to be dropped into various state-of-the-art NAS frameworks for NLP to accelerate the search time.
> * **Weakness 2:** Our evaluation setup of using machine translation benchmarks is borrowed from the NAS for NLP literature [Wang ACL'20, Jawahar ACL'23], which also uses only machine translation tasks. Our contribution is still non-trivial and significant, which as you’d pointed out is “innovative”, “valuable approach to reduce the cost of using LLMs for performance prediction”, “can be a substantial advantage for practitioners looking to optimize their deep learning models.”  Exploring other NLP tasks, computer vision, speech recognition tasks is orthogonal to our main research questions, but will be an important future direction.
> * **"Is the prediction result provided by LLM repeatable?"** We compute the predictions for 8500 randomly sampled architectures using LLM-PP GPT-4 three times and compute the standard deviation of the three predictions for each architecture. The mean of the standard deviation for 8500 architectures is very low: 0.21, 0.27, 0.27 BLEU for WMT’14 En-De, WMT’14 En-Fr, and WMT’19 En-De respectively. Thus, LLM-PP provides fairly robust performance predictions. For all our search experiments, we use a single estimate from LLM-PP.
> * **"...how do you ensure that LLM understands the DNN framework rather than reading memory..."** Below, we provide evidences on why the high performance of LLM-PP is highly unlikely due to “memorization”:
>   - Since the test set is created by randomly sampling from the search space, the probability of a test architecture to be part of the LLM training data is extremely low (e.g., 1 / 6000000 as the WMT’14 En-De search space has 6 million architectures). Note that we need a large number of resources (e.g.,  training a single architecture on 4 NVIDIA V100 GPUs takes 2 days) to train even a few 100s of architectures from scratch to convergence to get their true performance. Hence, it’s infeasible to have non-negligible coverage of architectures (along with its corresponding true performance) in the LLM training data.
>   - If the performance of LLM-PP is largely due to “memorization”, LLM-PP should achieve competitive performance even when the number of demonstrations in the prompt is minimal. Consider the table below.
>
> | Dataset | WMT’14 En-Fr | WMT’14 En-Fr | WMT’19 En-De | WMT’19 En-De |
> | - | ------ | ------ | ------ | ----- |
> | Perf. Predictor | MAE | Kendall-Tau | MAE | Kendall-Tau |
> | Neuron-wise MoS | 1.18 | 0.87 | 0.87 | 0.67 |
> | LLM-PP ChatGPT (June)  1 demonstration | 2.81 | 0.55 | 1.72 | 0.56 |
> | LLM-PP ChatGPT (June)  3 demonstrations |  2.28 | 0.67 | 1.12 | 0.57 |
> | LLM-PP ChatGPT (June) 10 demonstrations | 0.82 | 0.61 | 0.72 | 0.56 |
>
> From this table, it is clear that LLM-PP did not achieve competitive performance compared to the SOTA supernet baseline when the number of demonstrations in the prompt is minimal (e.g., less than or equal to 3 demonstrations). This result highlights that the performance of LLM-PP might not be largely due to “memorization”, and is likely due to other LLM capabilities such as “understanding” and “reasoning”. As the training data of LLMs like ChatGPT and GPT-4 are not public, it is impossible to conclusively say that the training data do not contain any of our test architectures. Despite this, we have shown strong evidence on why the performance of LLM-PP is highly unlikely to largely come from “memorization”.
> * **"why not use a PP that performs better than LLM-PP for distillation?..."** LLM-PP achieves the best performance (SOTA MAE, marginally worse Kendall-Tau but doesn’t have a big negative impact for the NAS task, as detailed in Section 5.2, second paragraph) compared to the supernet based performance predictors (see Table 1). Hence, LLM-PP-Distill is built by distilling the performance predictions from LLM-PP. In A.6.4, we show that LLM-PP-Distill that distills from HAT (marginally better Kendall-Tau than LLM-PP) leads to search hours reduction of at least 40% compared to HAT, but at the expense of big increase in latency and GFLOPs (9.7%, 32% respectively). This motivates why we distilled from LLM-PP.
>
> We hope we have answered your questions to your satisfaction and hope you are willing to raise the score. Please let us know if there are any further questions that we can address during the interactive rebuttal period.
>
> **References:**
> * [Wang ACL'20] HAT: Hardware-Aware Transformers for Efficient Natural Language Processing. Wang et al., ACL 2020.
> * [Jawahar ACL'23] AutoMoE: Heterogeneous Mixture-of-Experts with Adaptive Computation for Efficient Neural Machine Translation. Jawahar et al., ACL 2023.

---

> > ### Comment · Reviewer_ABPv · 2023-11-22
> > **Keep the score the same unless more results on nas benchmark are provided**
> >
> > Thanks for the author's response, which addressed most of my considerations.
> > For weakness 2, it is unfortunate that the authors did not validate the generalizability of the methodology on classical computer vision benchmarks, such as NAS-Bench-101/201.
> > As a result, my score remains above acceptable.

---

### Official Review · Reviewer_mogH · 2023-10-30

**Soundness:** 1 poor
**Presentation:** 3 good
**Contribution:** 1 poor
**Rating:** 3
**Confidence:** 4

**Summary:**

This paper uses GPT4 and few-shot learning with a specially designed prompt to predict model performance, and also employs a regression model trained on the distilled data to save costs (LLM-Distill-PP). Additionally, the paper proposes a hybrid search for NAS based on LLM-Distill-PP.

**Strengths:**

This paper presents an interesting method to predict model performance on a common model architecture, such as the transformer-base encoder-decoder version, and on a common dataset like WMT'14.

**Weaknesses:**

The effectiveness of the proposed method largely depends on how much information GPT-4 has "memorized." Since GPT-4 is a language model, its impressive prediction performance on WMT'14 (or WMT'19), transformer-base, translation direction, and BLEU is primarily because **these elements are commonly used for machine translation**. The authors need to recognize the limitations when dealing with less conventional models, datasets, translation directions, metrics, and other tasks and discuss these in the paper. For instance:

- What would occur if training and testing were done on WMT'22 data?
- What if the testing were on a low-resource language, say, Wolof?
- What would be the outcome when examining the results of COMET-22, or the recently released [COMET-kiwi-10B](https://huggingface.co/Unbabel/wmt23-cometkiwi-da-xxl) model, which GPT-4 lacks knowledge about?
- What if the chosen model were the [CNN-based embedding](https://arxiv.org/pdf/2305.14280.pdf) for machine translation, where GPT-4 has limited familiarity?

It's very likely that GPT has already encountered the architecture selection and results for your model, data, and metric settings since they have been prevalent in recent years. It is very possible that the author is **testing the model that it had been trained on the test dataset**. However, the author didn't explicitly address the performance in less conventional settings in the paper, rendering the study meaningless.

**Questions:**

Please see weaknesses above.

---

> ### Author Response · Authors · 2023-11-17
> **Response 1**
>
> Thanks for the careful review and for the helpful comments. We respond to quoted comments below:
> * **"The effectiveness of the proposed method largely depends on how much information GPT-4 has "memorized...."** Thank you for the questions. While WMT’14 and WMT’19 are popular MT benchmarks, it’s highly unlikely that the high performance of LLM-PP can be largely attributed to “memorization” of the underlying LLM for the following reasons:
>   - Since the test set is created by randomly sampling from the search space, the probability of a test architecture to be part of the LLM training data is extremely low (e.g., 1 / 6000000 as the WMT’14 En-De search space has 6 million architectures). Note that we need a large number of resources (e.g.,  training a single architecture on 4 NVIDIA V100 GPUs takes 2 days) to train even a few 100s of architectures from scratch to convergence to get their true performance. Hence, it’s infeasible to have non-negligible coverage of architectures (along with its corresponding true performance) in the LLM training data.
>   - If the performance of LLM-PP is largely due to “memorization”, LLM-PP should achieve competitive performance even when the number of demonstrations in the prompt is minimal. Consider the table below.
>
> | Dataset | WMT’14 En-Fr | WMT’14 En-Fr | WMT’19 En-De | WMT’19 En-De |
> | - | -------- | -------- | -------- | -------- |
> | Perf. Predictor | MAE | Kendall-Tau | MAE | Kendall-Tau |
> | Neuron-wise MoS | 1.18 | 0.87 | 0.87 | 0.67 |
> | LLM-PP ChatGPT (June)  1 demonstration | 2.81 | 0.55 | 1.72 | 0.56 |
> | LLM-PP ChatGPT (June)  3 demonstrations |  2.28 | 0.67 | 1.12 | 0.57 |
> | LLM-PP ChatGPT (June) 10 demonstrations | 0.82 | 0.61 | 0.72 | 0.56 |
>
> From this table, it is clear that LLM-PP did not achieve competitive performance compared to the SOTA supernet baseline when the number of demonstrations in the prompt is minimal (e.g., less than or equal to 3 demonstrations). This result highlights that the performance of LLM-PP might not be largely due to “memorization”, and is likely due to other LLM capabilities such as “understanding” and “reasoning”. Due to lack of time in the rebuttal phase, it might not be possible to experiment with other suggested datasets (we will send an update response if we could experiment with other datasets within the timeline).
> * **"It is very possible that the author is testing the model that it had been trained on the test dataset"** As the training data of LLMs like ChatGPT and GPT-4 are not public, it is impossible to conclusively say that the training data do not contain any of our test architectures. Despite this, we have shown strong evidence on why the performance of LLM-PP is highly unlikely to largely come from “memorization”.
>
> We hope we have answered your questions to your satisfaction and hope you are willing to raise the score. Please let us know if there are any further questions that we can address during the interactive rebuttal period.

---

> ### Comment · Reviewer_mogH · 2023-11-18
> **Memorization? Generalization?**
>
> I thank the authors for the response.
>
> I believe the authors should agree with the point that 'GPT-4 only works on the dataset (architecture information and its results) that it has seen before.' This is the fundamental reason why it also works on the common machine translation data (e.g., WMT14). This is the 'memorization' I am discussing. It predicts the model's performance based on what is most likely, with the restriction only on commonly used data (like WMT'14). That is also the reason I strongly disagree with the claim in the paper that 'LLMs have a ‘general understanding’ of DNN architectures.'
>
> The biggest flaw in this paper is that the authors fails to show the generalization of this method on uncommon data/architectures/languages (see my points in the initial review above). This method, under my understanding, is difficult to generalize and only applies to very commonly used data. Without showing the limitations on the uncommon data (or, say, generalization), this study should be a clear reject.

---

> > ### Author Response · Authors · 2023-11-23
> > **Generalization.**
> >
> > Thanks for your response.
> >
> > We disagree with the reviewer’s point that **”GPT-4 only works on the dataset (architecture information and its results) that it has seen before”**. Compared to SOTA performance predictors, LLM-PP GPT-4 works well for recent datasets (e.g., machine translation benchmark introduced in the 2023 year), low-resource/indigenous languages (e.g., Bribri, Chatino), and uncommon evaluation metrics (e.g., Comet), as discussed below.
> >
> > From the recent shared task: “AmericasNLP 2023 Shared Task on Machine Translation into Indigenous Languages” [AmericasNLP ACL’23], we take three machine translation benchmarks: Bribri to Spanish, Chatino to Spanish, and Spanish to Bribri. Compared to WMT 2014, WMT 2019 benchmarks (experimented in the paper), these three benchmarks are very recent (2023 year) and one of the languages in each translation direction is an low-resource/indigenous language (Bribri, Chatino). We compare LLM-PP GPT-4 against SOTA performance (BLEU) predictors on these benchmarks in terms of quality (MAE, Kendall-Tau). Consider the table below (performance averaged across two seeds),
> >
> > | | Bribri to Spanish | Bribri to Spanish | Chatino to Spanish | Chatino to Spanish | Spanish to Bribri | Spanish to Bribri|
> > |-|-|-|-|-|-|-|
> > |Performance Predictor | MAE | Kendall | MAE | Kendall | MAE | Kendall|
> > | HAT | 0.28 | 0.15 | 1.55 | **0.16** | 0.72  | 0.02 |
> > | Layer-wise MoS | 0.33 | -0.13 | 2.42 | -0.17 | 0.63 | -0.14 |
> > | Neuron-wise MoS | 0.29 | -0.35 | 2.94 | -0.06 | 0.43 | 0.09  |
> > | LLM-PP GPT-4 | **0.16** | **0.29** | **1.21** | 0.08 | **0.32** | **0.20** |
> >
> > It is clear that **LLM-PP achieves the SOTA MAE score across these benchmarks**, which is consistent with the trends in WMT 2014, WMT 2019 benchmarks (as shown in Table 1). Impressively, on two of these benchmarks, **LLM-PP also achieves the SOTA Kendall-Tau score**. Put together, these results clearly showcase that **LLM-PP generalizes well to recent datasets and low-resource languages.**
> >
> > Further, we show that LLM-PP also generalizes well to uncommon evaluation metrics. Based on the reviewer's suggestion, we build performance predictors that predict the Crosslingual Optimized Metric for Evaluation of Translation (COMET) [Comet HF] (Unbabel/wmt22-comet-da), which is relatively newer than the BLEU metric. Consider the table below (performance averaged across two seeds),
> >
> > | | Bribri to Spanish | Bribri to Spanish | Chatino to Spanish | Chatino to Spanish |
> > |-|-|-|-|-|
> > |Performance Predictor | MAE | Kendall | MAE | Kendall |
> > | HAT | 0.03 | 0.24 | 0.02 | -0.15 |
> > | Layer-wise MoS | 0.02 | -0.05 | 0.02 | 0.26 |
> > | Neuron-wise MoS | 0.02 | **0.32** | **0.01** | 0.34 |
> > | LLM-PP GPT-4 | **0.01** | **0.32** | **0.01** | **0.54** |
> >
> > On Bribri to Spanish task and Chatino to Spanish task, LLM-PP achieves the SOTA MAE and SOTA Kendall Tau performance compared to SOTA performance predictors. These results show that LLM-PP generalizes well to uncommon evaluation metrics like COMET. Note that we exclude Spanish to BriBri task, since COMET does not support Bribri.
> >
> > We will add these new results in the revision. Now, we will respond to the recent quoted comments below:
> > * **“'GPT-4 only works on the dataset ... that it has seen before.”** **”I strongly disagree with the claim in the paper that 'LLMs have a ‘general understanding’ of DNN architectures.'”** As shown above, we disagree with these claims as LLM-PP GPT-4 indeed generalizes to recent datasets, low-resource/indigenous language, and uncommon evaluation metrics, which are highly unlikely to be seen before by GPT-4,. These results can be largely attributed to LLM’s general understanding of DNN architectures.
> > * **”It predicts the model's performance based on what is most likely, with the restriction only on commonly used data (like WMT'14).”** **”This method, under my understanding, is difficult to generalize and only applies to very commonly used data.”** We disagree with these claims as LLM-PP GPT-4 generalizes to uncommon datasets such as Bribri to Spanish, Chatino to Spanish, and Spanish to Bribri, as shown above.
> > *  **”Without showing the limitations on the uncommon data (or, say, generalization), this study should be a clear reject.”** Thanks for the suggestions. In the current response, we showed that LLM-PP generalizes to machine translation datasets introduced in the 2023 year, low-resource/indigenous languages (Bribri, Chatino), and uncommon evaluation metrics (Comet). In the previous response, we already showed multiple pieces of evidence on why “memorization” cannot be a key factor for the superior performance of LLM-PP in general. We hope we have answered your questions to your satisfaction and hope you are willing to raise the score.
> >
> > Pls. let us know if there are any further questions.
> >
> > References
> > * [AmericasNLP ACL’23] https://turing.iimas.unam.mx/americasnlp/2023_st.html
> > * [Comet HF] https://huggingface.co/spaces/evaluate-metric/comet

---

### Official Review · Reviewer_oJTh · 2023-11-08

**Soundness:** 2 fair
**Presentation:** 2 fair
**Contribution:** 3 good
**Rating:** 5
**Confidence:** 4

**Summary:**

The authors design a new algorithm for neural architecture search that uses an LLM subroutine to predict the performance of neural architecture candidates (LLM-PP). They also introduce a modification which trains an MLP on the LLM-based predictions, to estimate architectures even beyond the ones that were predicted by the LLM (LLM-Distill-PP). The authors use an existing NAS framework with their LLM-Distill-PP method to perform NAS, which consists of using their method along with a supernetwork. The authors show several experiments on machine translation benchmarks.

**Strengths:**

Using LLMs for performance prediction is interesting and fairly novel.

Since LLMs are trained on the whole internet, with an emphasis on code, it is reasonable that an LLM would have an idea on the performance of architectures, especially well-known architectures.

The authors use the LLM-based supernet at the start of training, and then replace with a supernet. This fits the intuition that LLMs are strongest at performance prediction early on, but are no match for computational-based methods after a handful of iterations.

**Weaknesses:**

Overall, I am concerned that the paper is a bit too narrow in a few parts.

**Comparison to other methods.** The authors use three baselines, all of which are supernetwork-based performance predictors. The authors also make the statement, “The SOTA approach for building performance predictors (f_T ) is to train a weight-sharing supernet model on the task T.” It is highly unclear that this sentence is true. There are many different types of performance predictors, such as zero-cost proxies and learning curve extrapolation, each with different tradeoffs for runtime and accuracy. Furthermore, the performance of weight-sharing methods has been debated (e.g., the papers referenced here https://blog.ml.cmu.edu/2020/07/17/in-defense-of-weight-sharing-for-nas/).

I would have a better opinion of the experimental methodology if the authors compared to performance prediction methods beyond just supernetworks. Here are a few references:
- https://arxiv.org/abs/2008.03064
- https://proceedings.neurips.cc/paper_files/paper/2021/file/2130eb640e0a272898a51da41363542d-Paper.pdf
- https://proceedings.mlr.press/v188/laube22a/laube22a.pdf
- https://arxiv.org/abs/2101.08134

I would especially point out that recently, the extremely simple baseline, "number of parameters" [has been found](https://arxiv.org/abs/2008.03064) to be a surprisingly strong baseline for performance prediction, so this would be great to add as a baseline e.g. in Table 1.

Other than extending the set of baselines, I think the paper could be more impactful in other ways as well. For example, the authors only test their performance predictor on a single NAS framework; the one from HAT. There are other NAS frameworks too, for example, [Bayesian optimization](https://arxiv.org/abs/2110.10423) or BOHB.

Finally, the paper could also be more impactful if it tested search spaces / tasks beyond machine translation. For example, some of the above links use datasets based on computer vision, other NLP tasks, and speech recognition (like [this paper](https://arxiv.org/abs/2101.08134)).

It is surprising that LLM-Distill-PP performs better than LLM-PP. The authors give a few explanations. I think the paper would be stronger if the authors gave more insight and experiments into explaining this observation.

The authors mention that they share their code, but I couldn't find it. Can the authors share their code, e.g. with https://anonymous.4open.science/?

**Questions:**

If the authors can address some of the points in the weakness section, I would be open to raising my score. Specfiically, comparing to baselines would be important.

---

> ### Author Response · Authors · 2023-11-17
> **Response 1**
>
> Thanks for the careful review and for the helpful comments. We respond to quoted comments below:
> * **"I would especially point out that recently, the extremely simple baseline, "number of parameters" ... this would be great to add as a baseline e.g. in Table 1."**
> Thanks for these suggestions. The main reason to compare against supernetwork-based performance predictors only is that they are the state-of-the-art performance predictors in NAS for NLP literature, including machine translation (e.g., HAT [Wang ACL'20], AutoMoE [Jawahar ACL'23], Mixture-of-Supernets [Jawahar arXiv'23]) and BERT language modeling (e.g., NAS-BERT [Xu KDD'21], AutoDistil [Xu NeurIPS'22], Mixture-of-Supernets [Jawahar arXiv'23]). We will modify the SOTA statement in the revision as follows: “In NAS for NLP literature, the SOTA approach for building performance predictors (f_T ) is to train a weight-sharing supernet model on the task T.”
> Based on your suggestion, we experiment with the simple baseline “number of parameters”. The kendall-tau for this simple baseline is 0.42, 0.51, 0.54 for WMT’14 En-De, WMT’14 En-Fr, and WMT’19 En-De respectively, which is clearly worse than LLM-PP GPT-4 (0.65, 0.75, 0.65, as shown in Table 1 of the paper). Due to lack of time in the rebuttal phase, it might not be possible to experiment with other suggested baselines (we will send an update response if we could run other baselines within the timeline), but our work has compared to established SOTA performance predictors in NAS for NLP literature.
> * **".. the authors only test their performance predictor on a single NAS framework; the one from HAT. . There are other NAS frameworks too, for example, Bayesian optimization or BOHB."** We show the efficacy of HS-NAS in two SOTA NAS for MT frameworks: HAT and Mixture-of-Supernet (MoS) NAS framework. Due to lack of space, HS-NAS HAT results are in the main paper, while HS-NAS MoS results can be seen in Section A.6.3. Generalizing HS-NAS to other search frameworks such as bayesian optimization is orthogonal to our main research questions, but will be an important future direction.
> * **"Finally, the paper could also be more impactful if it tested search spaces / tasks beyond machine translation. ..."** Our evaluation setup of using machine translation benchmarks is borrowed from the NAS for NLP literature [Wang ACL'20, Jawahar ACL'23], which also uses only machine translation tasks. Our contribution is still non-trivial and significant, which as you’d pointed out is "interesting", "fairly novel", "fits the intuition that LLMs are strongest at performance prediction early on, but are no match for computational-based methods after a handful of iterations". Due to lack of time, other NLP tasks, computer vision, speech recognition cannot be included, but will be interesting future directions worth exploring.
> * **"It is surprising that LLM-Distill-PP performs better than LLM-PP. ..."** We believe there are two major reasons for this.
>   - LLM-Distill-PP is a linear regression model with only 486K parameters, while LLM-PP GPT-4 is a LLM with several billions of parameters. Due to smaller size, it is less likely that LLM-Distill-PP overfits compared to LLM-PP, thereby leading to better performance.
>   - LLM-Distill-PP is a specialist model as the model parameters are trained for performance prediction task with a few thousands of examples. On the other hand, LLM-PP is a generalist model which performs in-context learning with PP prompt and 10 demonstrations.
> * **"The authors mention that they share their code, but I couldn't find it. Can the authors share their code, e.g. with https://anonymous.4open.science/?"** Thanks for the suggestion. Please find the code in this link: https://anonymous.4open.science/r/llmpp-code-FF36
> * **"If the authors can address some of the points in the weakness section, I would be open to raising my score. Specifically, comparing to baselines would be important."** Thanks. We hope we have answered your questions to your satisfaction and hope you are willing to raise the score.
>
> Please let us know if there are any further questions that we can address during the interactive rebuttal period.
>
> **References:**
> * [Wang ACL'20] HAT: Hardware-Aware Transformers for Efficient Natural Language Processing. Wang et al., ACL 2020.
> * [Jawahar ACL'23] AutoMoE: Heterogeneous Mixture-of-Experts with Adaptive Computation for Efficient Neural Machine Translation. Jawahar et al., ACL 2023.
> * [Jawahar arXiv'23] Mixture-of-Supernets: Improving Weight-Sharing Supernet Training with Architecture-Routed Mixture-of-Experts. Jawahar et al., arXiv:2306.04845, 2023.
> * [Xu KDD'21] NAS-BERT: Task-Agnostic and Adaptive-Size BERT Compression with Neural Architecture Search. Xu et al., KDD 2021.
> * [Xu NeurIPS'22] AutoDistil: Few-shot Task-agnostic Neural Architecture Search for Distilling Large Language Models. Xu et al., NeurIPS 2022.

---

> > ### Comment · Reviewer_oJTh · 2023-11-22
> >
> > Thank you for the response. Overall I have a better view of the paper.
> >
> > It is great that you added “number of parameters” as a baseline. I agree that it is tough to add during the short author response period, but I still think that comparing to other methods would improve the evaluation, since the NLP for NAS community may not necessarily always include a diverse set of performance predictors in all papers, and their performance can be versatile depending on the desired training time, inference time, and NAS algorithm.
> >
> > Thank you for clarifying that there is another NAS algorithm, HS-NAS MoS in Appendix A.6.3.
> >
> > Again, I appreciate the clarification that it is standard in the NLP NAS community. Although, I think the paper could still have more reach if it covered other search spaces and tasks. Unless I am mistaken, there is nothing specific to NLP in the method. I also tend to agree with Reviewer mogH, that the paper could especially benefit from additional tasks that have less risk of “training data contamination” of the LLM.
> >
> > “Due to smaller size, it is less likely that LLM-Distill-PP overfits” -> yes, actually, it is likely that using an MLP acts as a regularizer for the raw LLM predictions.
> >
> > Thank you for including the code!
> >
> > Based on the above strengths, I am raising my score, although I think the paper could still be better if it e.g. includes more recent tasks.

---

> > > ### Author Response · Authors · 2023-11-23
> > > **More non-supernet baselines. Added recent tasks.**
> > >
> > > Thanks for your response and raising your score. We respond to the new quoted comments below:
> > >
> > > * **”It is great that you added “number of parameters” as a baseline. I agree that it is tough to add during the short author response period, but I still think that comparing to other methods would improve the evaluation, ….”** Thanks for the suggestions. We add comparison to **five** non-supernet baselines: #Params, #FLOPs, grad_norm, snip, and snyflow (see [Zero-cost ICLR’22 Blog] for details). Consider the table below,
> > >
> > > | Kendall | WMT’14 En-De | WMT’14 En-Fr | WMT’19 En-De |
> > > | - | - | - | - |
> > > | #Params | 0.42 | 0.51| 0.54 |
> > > | #FLOPs | 0.43 | 0.53 | 0.54 |
> > > | grad_norm | -0.42 | -0.42 | -0.52 |
> > > | snip | -0.42 | -0.27 | -0.3 |
> > > |synflow | -0.31 | -0.47 | -0.49 |
> > > |LLM-PP GPT-4 | **0.65** | **0.75** | **0.65** |
> > >
> > > It is clear that LLM-PP GPT-4 achieves a high Kendall Tau, outperforming all the non-supernet baselines. These results along with Table 1 from the paper showcases the superior performance of LLM-PP across a wide range of baselines.
> > > * **”I also tend to agree with Reviewer mogH, that the paper could especially benefit from additional tasks that have less risk of “training data contamination” of the LLM.”** Thanks for the suggestions. Compared to SOTA performance predictors, LLM-PP GPT-4 works well for recent datasets (e.g., 2023 benchmark), low-resource/indigenous languages (e.g., Bribri, Chatino), and uncommon evaluation metrics (e.g., Comet), as discussed below.
> > >
> > > From the recent shared task: “AmericasNLP 2023 Shared Task on Machine Translation into Indigenous Languages” [AmericasNLP ACL’23], we take three machine translation benchmarks: Bribri to Spanish, Chatino to Spanish, and Spanish to Bribri. Compared to WMT 2014, WMT 2019 benchmarks (experimented in the paper), these three benchmarks are very recent (2023 year) and one of the languages in each translation direction is an low-resource/indigenous language (Bribri, Chatino). We compare LLM-PP GPT-4 against SOTA performance (BLEU) predictors on these benchmarks in terms of quality (MAE, Kendall-Tau). Consider the table below (performance averaged across two seeds),
> > >
> > > | | Bribri to Spanish | Bribri to Spanish | Chatino to Spanish | Chatino to Spanish | Spanish to Bribri | Spanish to Bribri|
> > > |-|-|-|-|-|-|-|
> > > |Performance Predictor | MAE | Kendall | MAE | Kendall | MAE | Kendall|
> > > | HAT | 0.28 | 0.15 | 1.55 | **0.16** | 0.72  | 0.02 |
> > > | Layer-wise MoS | 0.33 | -0.13 | 2.42 | -0.17 | 0.63 | -0.14 |
> > > | Neuron-wise MoS | 0.29 | -0.35 | 2.94 | -0.06 | 0.43 | 0.09  |
> > > | LLM-PP GPT-4 | **0.16** | **0.29** | **1.21** | 0.08 | **0.32** | **0.20** |
> > >
> > > It is clear that **LLM-PP achieves the SOTA MAE score across these benchmarks**, which is consistent with the trends in WMT 2014, WMT 2019 benchmarks (as shown in Table 1). Impressively, on two of these benchmarks, **LLM-PP also achieves the SOTA Kendall-Tau score**. Put together, these results clearly showcase that **LLM-PP generalizes well to recent datasets and low-resource languages.**
> > >
> > > Further, we show that LLM-PP also generalizes well to uncommon evaluation metrics. Based on the reviewer's suggestion, we build performance predictors that predict the Crosslingual Optimized Metric for Evaluation of Translation (COMET) [Comet HF] (Unbabel/wmt22-comet-da), which is relatively newer than the BLEU metric. Consider the table below (performance averaged across two seeds),
> > >
> > > | | Bribri to Spanish | Bribri to Spanish | Chatino to Spanish | Chatino to Spanish |
> > > |-|-|-|-|-|
> > > |Performance Predictor | MAE | Kendall | MAE | Kendall |
> > > | HAT | 0.03 | 0.24 | 0.02 | -0.15 |
> > > | Layer-wise MoS | 0.02 | -0.05 | 0.02 | 0.26 |
> > > | Neuron-wise MoS | 0.02 | **0.32** | **0.01** | 0.34 |
> > > | LLM-PP GPT-4 | **0.01** | **0.32** | **0.01** | **0.54** |
> > >
> > > On Bribri to Spanish task and Chatino to Spanish task, LLM-PP achieves the SOTA MAE and SOTA Kendall Tau performance compared to SOTA performance predictors. These results show that LLM-PP generalizes well to uncommon evaluation metrics like COMET. Note that we exclude Spanish to BriBri task, since COMET does not support Bribri.
> > > * **”Based on the above strengths, I am raising my score, although I think the paper could still be better if it e.g. includes more recent tasks.”** Thanks for these suggestions. As shown above, we have added more recent tasks and showed that LLM-PP generalizes to machine translation datasets introduced in the 2023 year, low-resource/indigenous languages (Bribri, Chatino), and uncommon evaluation metrics (Comet).
> > >
> > > We will add these new results in the revision. Overall, we hope we have answered your questions to your satisfaction and hope you are willing to raise the score further. Pls. let us know if there are any further questions.
> > >
> > >
> > > References:
> > > * [Zero-cost ICLR’22 Blog] https://iclr-blog-track.github.io/2022/03/25/zero-cost-proxies/
> > > * [AmericasNLP ACL’23] https://turing.iimas.unam.mx/americasnlp/2023_st.html
> > > * [Comet HF] https://huggingface.co/spaces/evaluate-metric/comet

---

### Official Review · Reviewer_GzhX · 2023-11-11

**Soundness:** 3 good
**Presentation:** 1 poor
**Contribution:** 3 good
**Rating:** 6
**Confidence:** 3

**Summary:**

The paper presents an innovative approach to using Large Language Models (LLMs) for building performance predictors (PP). The authors have designed PP prompts for LLMs and demonstrated that GPT-4, when equipped with these prompts, can predict the performance of architectures with significant accuracy. The paper further introduces a distilled regression model, LLM-Distill-PP, and proposes a hybrid search algorithm for Neural Architecture Search (NAS), demonstrating its efficiency and potential.

**Strengths:**

- Innovative use of LLMs for the purpose of performance prediction.
- The introduction of LLM-Distill-PP and the Hybrid-Search algorithm significantly reduces the latency in searching for architectures.
- Extensive experiments demonstrate the efficiency of the proposed methods, highlighting their practicality.

**Weaknesses:**

- The paper could benefit from a more in-depth exploration of the validation methods used. The explanations in Sections 3 and 4 do not clearly articulate the problem statement and baseline comparisons.
- While the concept of distillation is critical, the paper's narrative feels disjointed. The scientific discourse between Chapters 5 and 6 appears fragmented and could be more cohesively presented.
- The figures require refinement; the font aesthetics are lacking, particularly in Figure 2. Algorithm 1 needs redesigning for better readability. The overall structure of the paper could be improved for clarity and flow.

**Questions:**

- How does the LLM-Distill-PP model's efficiency and accuracy compare to other existing models?
- Could the authors elaborate on the rationale and design process behind the PP prompts used for LLMs?
- Is the proposed Hybrid-Search algorithm scalable for larger datasets or more complex architectures, and if so, how?

---

> ### Author Response · Authors · 2023-11-17
> **Response**
>
> Thanks for the careful review and for the helpful comments. We will improve the presentation aspects of the paper as per your suggestions in the revision. We respond to quoted comments below:
> * **"How does the LLM-Distill-PP model's efficiency and accuracy compare to other existing models?"** The following table shows the efficiency (time taken to predict performance for 10 architectures) and accuracy (MAE, Kendall-Tau, as shown in Table 1) for supernet-based PP (HAT, Layer-wise MoS, Neuron-wise MoS), LLM-PP (GPT-4), and LLM-Distill-PP (GPT-4) for the WMT'14 En-De task.
> | Perf. Predictor (PP)| MAE | Kendall-Tau | Prediction Time (seconds) |
> | -- | -- | -- | -- |
> | HAT | 1.14 | 0.71 | 10.5 |
> | Layer-wise MoS | 1.05 | 0.81 | 13.9 |
> | Neuron-wise MoS | 0.97  | 0.56 | 13.3 |
> | LLM-PP GPT-4 | 0.28 | 0.65 | 11.9 |
> | LLM-Distill-PP GPT-4  | 0.22 | 0.64 | 0.01|
>
> It is clear that LLM-Distill-PP provides the best efficiency-accuracy tradeoff with on par (if not better) accuracy as LLM-PP but significantly faster prediction time (0.01s vs. 11.9s).
> * **"Could the authors elaborate on the rationale and design process behind the PP prompts used for LLMs?"** The goal of creating PP prompts is to precisely communicate the performance prediction task to the LLM. We selected machine translation benchmarks due to their extensive study in the NAS for NLP literature. The design process began by examining crucial elements of the machine translation task, commonly used model architectures, and relevant efficiency metrics. Initially, we presented only demonstrations, borrowing hyperparameter wording from HAT’s configuration file. Subsequently, we added the role and definition of each hyperparameter, using wording from HAT’s helper description. Moving forward, our aim was to craft instructions enabling the LLM to grasp essential tasks, architecture, and metric details. Most instructions are prefixed with 'You should' to encourage strict adherence. Five instructions were incorporated. The first specifies the dataset, translation direction, and quality metric. The second provides examples randomly sampled from the training set, presented with generic prefixes ('Input:' for source sentence, 'Output:' for target sentence). The third outlines the architecture, citing the 'Attention Is All You Need' paper, assuming the LLM is familiar with this popular work. Standard settings and optimization algorithms are noted for training the architectures. The fourth identifies the efficiency metric in the demonstrations. The final instruction aims to summarize the relationships the LLM should learn to solve the task effectively.
> * **"Is the proposed Hybrid-Search algorithm scalable for larger datasets or more complex architectures, and if so, how?"** Thanks for the question. By design, the search algorithms (hybrid-search, baseline search) are scalable to: (i) larger datasets as the search time is agnostic to the size of the dataset and depends only on the hyperparameter settings (e.g., num-iterations, num-mutations, num-crossover, num-parents) of the search algorithm, and (ii) complex architectures as long as the PP prompts are adapted (e.g., third instruction in the PP prompt should contain the details of the complex architecture, and demonstrations in PP prompt contain the hyperparameter values of that architecture).
>
> We hope we have answered your questions to your satisfaction and hope you are willing to raise the score. Please let us know if there are any further questions that we can address during the interactive rebuttal period.

---

### Meta-Review · Area_Chair_hnbM · 2023-12-06

**Metareview:**

This is a timely paper on using LLMs for NAS performance prediction based on in-context presentation of architectures.
The paper focusses narrowly on NAS applications for machine translation, and as such uses what for general NAS researchers are a very limited number of baselines. I do encourage the authors to look into the literature recommended by the reviewers, as well as the 31 performance predictors in https://arxiv.org/abs/2104.01177, to see which ones could also be used. E.g., the LLM does receive architecture evaluations as input, and the same evaluations should be possible to feed to Bayesian optimization.
Overall, the paper is very promising, but the empirical evaluation has some issues. I encourage the authors to include additional baselines in the future. Review scores of 3,5,6,6 put the paper in the rejection zone, and in the private reviewer-AC discussion, none of the reviewers spoke up for the paper. I therefore recommend rejection but encourage the reviewers to continue this pirv

**Justification For Why Not Higher Score:**

The set of baselines for the empirical evaluation is incomplete.

**Justification For Why Not Lower Score:**

N/A

---

### Decision · Program_Chairs · 2024-01-16

Reject